# Online Continual Learning on Class Incremental Blurry Task Configuration with Anytime Inference

**Hyunseo Koh**[1,3,*]   **Dahyun Kim**[2,3,*]   **Jung-Woo Ha**[3]   **Jonghyun Choi**[3,4,†]
[1]GIST, South Korea   [2]Upstage AI Research   [3]NAVER AI Lab.   [4]Yonsei University
`hyunseo8157@gm.gist.ac.kr, kdahyun@upstage.ai`
`jungwoo.ha@navercorp.com, jc@yonsei.ac.kr`

## Abstract

Despite rapid advances in continual learning, a large body of research is devoted to improving performance in the existing setups. While a handful of work do propose new continual learning setups, they still lack practicality in certain aspects. For better practicality, we first propose a novel continual learning setup that is online, task-free, class-incremental, of blurry task boundaries and subject to inference queries at any moment. We additionally propose a new metric to better measure the performance of the continual learning methods subject to inference queries at any moment. To address the challenging setup and evaluation protocol, we propose an effective method that employs a new memory management scheme and novel learning techniques. Our empirical validation demonstrates that the proposed method outperforms prior arts by large margins. Code and data splits are available at `https://github.com/naver-ai/i-Blurry`.

## 1 Introduction

Continual learning (CL) is a learning scenario where a model learns from a continuous and online stream of data and is regarded as a more realistic and practical learning setup than offline learning on a fixed dataset (He et al., 2020). However, many CL methods still focus on the offline setup (Kirkpatrick et al., 2017; Rebuffi et al., 2017; Saha et al., 2021) instead of the more realistic online setup. These methods assume access to a large storage, storing the entire data of the current task and iterating on it multiple times. On the other hand, we are interested extensively in the more realistic online setup where only a small memory is allowed as storage. Meanwhile, even for the online CL methods, we argue they have room for more practical and realistic improvements concerning multiple crucial aspects. The aspects include the class distributions such as the disjoint (Rebuffi et al., 2017) or the blurry (Aljundi et al., 2019c) splits and the evaluation metric that focuses only on the task accuracy such as average task accuracy ($A_{avg}$).

The two main assumptions on the class distributions in existing CL setups, *i.e.*, the disjoint and blurry splits, are less realistic for the following reasons. The disjoint split assumes no classes overlap over different tasks; already observed classes will never appear again.

The above is not plausible because already observed classes can still appear later on in real-world scenarios (see Fig. 2 of (Bang et al., 2021)). On the other hand, in the blurry split (Aljundi et al., 2019c) no new classes appear after the first task even though the split assumes overlapping classes over tasks. This is also not plausible as observing new classes is common in real-world scenarios.

The typical evaluation metric such as $A_{avg}$ in which the accuracy is measured only at the task transition is also less realistic. It implicitly assumes that no inference queries occur in the middle of a task. However, in real-world scenarios, inference queries can occur *at any-time*. Moreover, there is no explicit task transition boundary in most real-world scenarios. Thus, it is desirable for CL

---

* indicates equal contribution.

† indicates corresponding author.

This work was done while HK, DK and JC were interns and an AI technical advisor at NAVER AI Lab.

models to provide good inference results at any time. To accurately evaluate whether a CL model is effective at such 'any-time' inference, we need a new metric for CL models.

In order to address the issues of the current CL setups, we propose a new CL setup that is more realistic and practical by considering the following criteria: First, the class distribution is comprised of the advantages from both blurry and disjoint. That is, we assume that the model continuously encounters new classes as tasks continue, *i.e.*, class-incremental and that classes overlap across tasks, *i.e.*, blurry task boundaries, while not suffering from the restrictions of blurry and disjoint. Second, the model is evaluated throughout training and inference such that it can be evaluated for any-time inference. We call this new continual learning setup **'i-Blurry'**.

For the i-Blurry setup, we first propose a plausible baseline using experience replay (ER) with reservoir sampling and a tuned learning rate scheduling. While existing online CL methods are applicable to the i-Blurry setup, they perform only marginally better than our baseline or often worse.

To better handle the i-Blurry setup, we propose a novel continual learning method, which improves the baseline in three aspects. We design a new memory management scheme to discard samples using a per-sample importance score that reflects how useful a sample is for training. We then propose to draw training samples only from the memory instead of drawing them from both memory and the online stream as is done in ER. Finally, we propose a new learning rate scheduling to adaptively decide whether to increase or decrease the learning rate based on the loss trajectory, *i.e.* a data-driven manner. To evaluate the algorithms in the new setup, we evaluate methods by conventional metrics, and further define a new metric called 'area under the curve of accuracy' ($A_{\text{AUC}}$) which measures the model's accuracy throughout training.

We summarize our contributions as follows:

- Proposing a new CL setup called i-Blurry, which addresses a more realistic setting that is online, task-free, class-incremental, of blurry task boundaries, and subject to any-time inference.
- Proposing a novel online and task-free CL method by a new memory management, memory usage, and learning rate scheduling strategy.
- Outperforming existing CL models by large margins on multiple datasets and settings.
- Proposing a new metric to better measure a CL model's capability for the desirable any-time inference.

## 2 RELATED WORK

**Continual learning setups.** There are many CL setups that have been proposed to reflect the real-world scenario of training a learning model from a stream of data (Prabhu et al., 2020). We categorize them in the following aspects for brevity.

First, we categorize them into (1) task-incremental (*task-IL*) and (2) class-incremental learning (*class-IL*), depending on whether the task-ID is given at test time. Task-IL, also called multi-head setup, assumes that task-ID is given at test time (Lopez-Paz & Ranzato, 2017; Aljundi et al., 2018; Chaudhry et al., 2019). In contrast, in class-IL, or single-head setup, task-ID is not given at test time and has to be inferred (Rebuffi et al., 2017; Wu et al., 2019; Aljundi et al., 2019a). Class-IL is more challenging than task-IL, but is also more realistic since task-ID will not likely be given in the real-world scenario (Prabhu et al., 2020). Most CL works assume that task ID is provided at training time, allowing CL methods to utilize the task ID to save model parameters at task boundaries (Kirkpatrick et al., 2017; Chaudhry et al., 2018b) for later use. However, this assumption is impractical (Lee et al., 2019) since real-world data usually do not have clear task boundaries. To address this issue, a task-free setup (Aljundi et al., 2019b), where task-ID at training is not available, has been proposed. We focus extensively on the task-free setup as it is challenging and being actively investigated recently (Kim et al., 2020; Lee et al., 2019; Aljundi et al., 2019c).

We now categorize CL setups into *disjoint* and *blurry* setup by how the data split is configured. In the disjoint task setup, each task consists of a set of classes disjoint from all other tasks. But the disjoint setup is less realistic as the classes in the real-world can appear at any time not only in a disjoint manner. Recently, to make the setup more realistic, a blurry task setup has been proposed and investigated (Aljundi et al., 2019c; Prabhu et al., 2020; Bang et al., 2021), where $100 - M\%$ of the sampels are from the dominant class of the task and $M\%$ of the samples are from all classes,

where $M$ is the blurry level (Aljundi et al., 2019c). However, the blurry setup assumes *no class is added* in new tasks, *i.e.*, not class-incremental, which makes the setup still not quite realistic.

Finally, depending on how many samples are streamed at a time, we categorize CL setups into *online* (Rolnick et al., 2018; Aljundi et al., 2019a; Chaudhry et al., 2019) and *offline* (Wu et al., 2019; Rebuffi et al., 2017; Chaudhry et al., 2018b; Castro et al., 2018). In the offline setup, all data from the current task can be used an unlimited number of times. This is impractical since it requires additional memory of size equal to the current task's data. For the online setup, there are many notions of 'online' that differs in each literature. Prabhu et al. (2020); Bang et al. (2021) refer online to a setup using each streamed sample *only once* to train a model while Aljundi et al. (2019c;a) refer online to a setup where only one or a few samples are streamed at a time. We follow the latter as the former allows storing the whole task's data, which is similar to offline and less realistic.

In this work, we propose a novel CL setup that is online, task-free, class-incremental, of blurry task boundaries, and subject to any-time inference as the most realistic setup for continual learning. Note that task-free and class-incremental are compatible (Jin et al., 2020; van de Ven et al., 2021).

**Continual learning methods.** Given neural networks would suffer from catastrophic forgetting (McCloskey & Neal, 1989; Ratcliff, 1990), the online nature of streaming data in continual learning generally aggravates the issue. To alleviate the forgetting, there are various proposals to store the previous task information; (1) regularization, (2) replay, and (3) parameter isolation.

(1) Regularization methods (Kirkpatrick et al., 2017; Zenke et al., 2017; Lee et al., 2017b; Ebrahimi et al., 2020) store previous task information in the form of model priors and use it for regularizing the neural network currently being trained. (2) Replay methods store a subset of the samples from the previous tasks in an *episodic memory* (Rebuffi et al., 2017; Castro et al., 2018; Chaudhry et al., 2019; Wu et al., 2019; Kim et al., 2019) or keep a generative model that is trained to generate previous task samples (Shin et al., 2017; Wu et al., 2018; Hu et al., 2019; Cong et al., 2020). The sampled or generated examplars are replayed on future tasks and used for distillation, constrained training, or joint training. (3) Parameter isolation methods augment the networks (Rusu et al., 2016; Lee et al., 2017a; Aljundi et al., 2017) or decompose the network into subnetworks for each task (Mallya & Lazebnik, 2018; Cheung et al., 2019; Yoon et al., 2020).

Since (1), (2), and (3) all utilize different ways of storing information that incurs parameter storage costs, episodic memory requirement and increase in network size respectively, a fair comparison among the methods is not straighforward. We mostly compare our method with episodic memory-based methods (Aljundi et al., 2019a; Prabhu et al., 2020; Bang et al., 2021) due to performance, but also with methods that use regularization as well (Chaudhry et al., 2018b; Wu et al., 2019).

**Online continual learning.** Despite being more realistic (Losing et al., 2018; He et al., 2020), online CL have not been popular (Prabhu et al., 2020) due to the difficulty and subtle differences in the setups in the literature. ER (Rolnick et al., 2018) is a simple yet strong episodic memory-based online CL method using reservoir sampling for memory management and jointly trains a model with half of the batch sampled from memory. Many online CL methods are based on ER (Aljundi et al., 2019c;a). GSS (Aljundi et al., 2019c) selects samples based on cosine similarity of gradients. MIR (Aljundi et al., 2019a) retrieves maximally interfering samples from memory to use for training.

Different from ER, A-GEM (Chaudhry et al., 2019) uses the memory to enforce constraints on the loss trajectory of the stored samples. GDumb (Prabhu et al., 2020) only updates the memory during training phase and trains from scratch at the test time only using the memory.

The recently proposed RM (Bang et al., 2021) uses an uncertainty-based memory sampling and two-stage training scheme where the model trains for one epoch on the streamed samples and trains extensively only using the memory at the end of each task, delaying most of the learning to the end of each task. The uncertainty-based memory sampling is not well-suited for online CL and the two-stage training leads to poor any-time inference. Our method outperforms all online CL methods introduced in this section while strictly adhering to the online and task-free restrictions.

## 3  PROPOSED CONTINUAL LEARNING SETUP: I-BLURRY

For a more realistic and practical CL setup, considering real-world scenarios, we strictly adhere to the online and task-free CL setup (Lee et al., 2019; Losing et al., 2018). Specifically, we propose

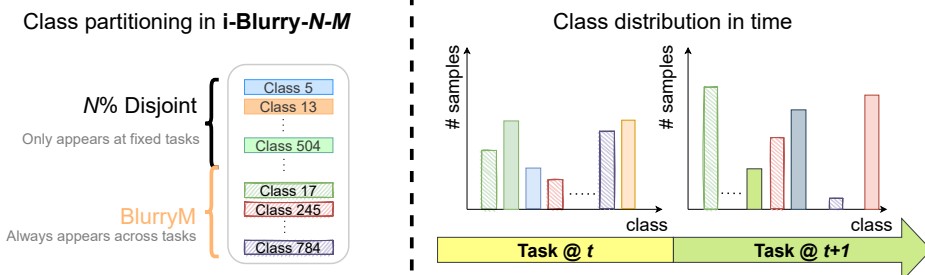

Figure 1: i-Blurry-$N$-$M$ split. $N\%$ of classes are partitioned into the disjoint set and the rest into the Blurry$M$ set where $M$ denotes the blurry level (Aljundi et al., 2019c). To form the i-Blurry-$N$-$M$ task splits, we draw training samples from a uniform distribution from the 'disjoint' or the 'Blurry$M$' set (Aljundi et al., 2019c). The 'blurry' classes always appear over the tasks while disjoint classes gradually appear.

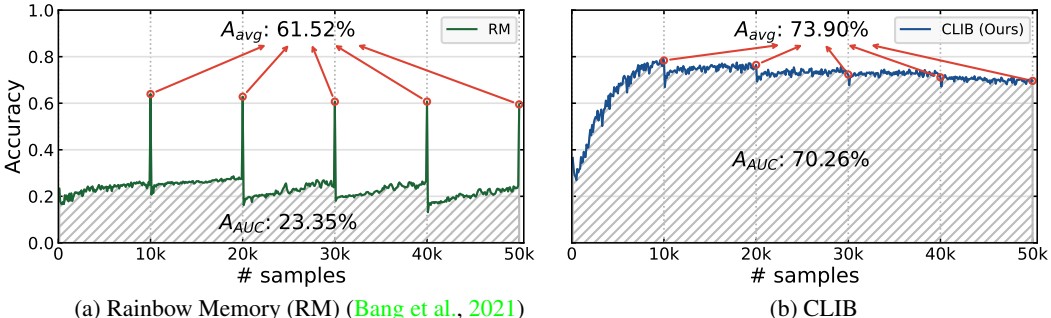

(a) Rainbow Memory (RM) (Bang et al., 2021)     (b) CLIB

Figure 2: Comparison of $A_{\text{AUC}}$ with $A_{avg}$. (a) online version of RM (Bang et al., 2021) (b) proposed CLIB. The two-stage method delays most of the training to the end of the task. The accuracy-to-{# of samples} plot shows that our method is more effective at any time inference than the two-stage method. The difference in $A_{avg}$ for the two methods is much smaller than that of in $A_{\text{AUC}}$, implying that $A_{\text{AUC}}$ captures the effectiveness at any-time inference better.

a novel CL setup (named as **i-Blurry**), with two characteristics: 1) class distribution being class incremental and having blurry task boundaries and 2) allowing for any-time inference.

**i-Blurry-N-M Split.** We partition the classes into groups where $N\%$ of the classes are for disjoint and the rest of $100-N\%$ of the classes are used for Blurry$M$ sampling (Aljundi et al., 2019c), where $M$ is the blurry level. Once we determine the partition, we draw samples from the partitioned groups. We call the resulting sequence of tasks as **i-Blurry-$N$-$M$** split. The i-Blurry-$N$-$M$ splits feature both class-incremental and blurry task boundaries. Note that the i-Blurry-$N$-$M$ splits generalize previous CL setups. For instance, $N = 100$ is the disjoint split as there are no blurry classes. $N = 0$ is the Blurry$M$ split (Aljundi et al., 2019c) as there are no disjoint classes. $M = 0$ is the disjoint split as the blurry level is $M = 0$ (Bang et al., 2021). We use multiple i-Blurry-$N$-$M$ splits for reliable empirical validations and share the splits. Fig. 1 illustrates the i-Blurry-$N$-$M$ split.

**A New Metric – Area Under the Curve of Accuracy** ($A_{\text{AUC}}$). Average accuracy (*i.e.*, $A_{avg} = \frac{1}{T}\sum_{i=1}^{T} A_i$ where $A_i$ is the accuracy at the end of the i$^{\text{th}}$ task) is one of the widely used measures in continual learning. But $A_{avg}$ only tells us how good a CL model is at the few discrete moments of task transitions ($5 - 10$ times for most CL setups) when the model could be queried at any time. Thus, a CL method could be poor at any-time inference but the $A_{avg}$ may be insufficient to deduce that conclusion due to its temporal sparsity of measurement. For example, Fig. 2 compares the online version of RM (Bang et al., 2021), which conducts most of the training by iterating over the memory at the end of a task, with our method. RM is shown as it is a very recent method that performs well on $A_{avg}$ but particularly poor at any-time inference. Only evaluating with $A_{avg}$ might give the false sense that the difference between the two methods is not severe. However, the accuracy-to-{# of samples} curve reveals that our method shows much more consistently high accuracy during training, implying that our method is more suitable for any-time inference than RM. To alleviate the limitations of $A_{avg}$, we shorten the accuracy measuring frequency to after every $\Delta n$

Figure 3: Overview of the proposed CLIB. We compute sample-wise importance during training to manage our memory. Note that we only draw training samples from the memory whereas ER based methods draw them from both the memory and the online stream.

samples are observed instead of at discrete task transitions. The new metric is equivalent to the area under the curve (AUC) of the accuracy-to-{# of samples} curve for CL methods when $\Delta n = 1$. We call it area under the curve of accuracy ($A_{\text{AUC}}$):

$$A_{\text{AUC}} = \sum_{i=1}^{k} f(i \cdot \Delta n) \cdot \Delta n, \tag{1}$$

where the step size $\Delta n$ is the number of samples observed between inference queries and $f(\cdot)$ is the curve in the accuracy-to-{# of samples} plot. High $A_{\text{AUC}}$ corresponds to a CL method that consistently maintains high accuracy throughout training.

The large difference in $A_{\text{AUC}}$ (see Fig. 4) implies that delaying strategies like the two-stage training scheme are not effective for any-time inference, a conclusion harder to deduce with just $A_{avg}$.

## 4 METHOD

### 4.1 A BASELINE FOR I-BLURRY SETUP

To address the realistic i-Blurry setup, we establish a baseline for the challenging online and task-free i-Blurry setup. For the memory management policy, we use reservoir sampling (Vitter, 1985) and for memory usage, we we use experience replay (ER). For the LR scheduling, we use an exponential LR schedule but reset the LR when a new class is encountered. Please see Sec. A.1 for details.

Note that the above baseline still has room to improve. The reservoir sampling does not consider whether one sample could be more useful for training than the others. ER uses samples from the stream directly, which can skew the training of CL models to recently observed samples. While the exponential with reset does increase the LR periodically, the sudden changes may disrupt the CL model. Thus, we discuss how we can improve the baseline in the following sections. The final method with all the improvements is illustrated in Fig. 3.

### 4.2 SAMPLE-WISE IMPORTANCE BASED MEMORY MANAGEMENT

In reservoir sampling, the samples are removed from the memory at random. Inspired by works on sample importance (Kloek & Van Dijk, 1978; LeCun et al., 1990; Chang et al., 2017; Katharopoulos & Fleuret, 2018; Csiba & Richtárik, 2018), we propose a novel sampling strategy specific for CL that removes samples from the memory based on sample-wise importance as following Theorem 1.

**Theorem 1.** *Let $\mathcal{C} = \mathcal{M} \cup \{(x_{new}, y_{new})\}$, $\mathcal{M}$ be a memory from the previous time step, $(x_{new}, y_{new})$ be the newly encountered sample, $l(\cdot)$ be the loss and $\theta$ be the model parameters. Assuming that the model trained with the optimal memory, $\mathcal{M}^*$ will induce maximal loss decrease on $\mathcal{C}$, the optimal memory is given by $\mathcal{M}^* = \mathcal{C} \setminus \{(\bar{x}, \bar{y})\}$ with*

$$(\bar{x}, \bar{y}) = \underset{(x_i, y_i) \in \mathcal{C}}{\arg \min} \, \mathbb{E}_{\theta} \left[ \sum_{(x,y) \in \mathcal{C}} l(x, y; \theta) - l\left(x, y; \theta - \nabla_{\theta} l(x_i, y_i; \theta)\right) \right]. \tag{2}$$

*Proof.* Please see Sec. A.2 for the proof. □

Theorem 1 states that when a new sample is appended to the memory and a sample has to be discarded, we should discard the sample that incurs the least loss decrease *i.e.*, least useful for training. We solve Eq. 2 by keeping track of the sample-wise importance $\mathcal{H}_i$:

$$\mathcal{H}_i = \mathbb{E}_\theta \left[ \sum_{(x,y) \in \mathcal{C}} l(x, y; \theta) - l\left(x, y; \theta - \nabla_\theta l(x_i, y_i; \theta)\right) \right]. \tag{3}$$

Intuitively, $\mathcal{H}$ is the expected loss decrease when the associated sample is used for training. We update $\mathcal{H}$ associated with the samples used for training after every training iteration. Specifically, the model is trained with a random batch from the memory. Then, for all samples in $C$, we measure the loss difference before and after training with the batch. If the loss decreases, the importance scores of the samples in the batch increase and vice versa. Note that because the importance score of a sample is relative to that of other samples in the memory, when the importance scores of the samples in the batch decrease, the scores of the samples not in the batch increase comparatively. Thus, two types of samples have high importance scores; 1) samples that were in the batch when the loss decreased and 2) samples that were not in the batch when the loss increased.

The expectation in Eq. 3 is taken over $\theta$'s optimization trajectory *i.e.*, over time. For computational efficiency, we use the exponential moving average as empirical estimates instead. The empirical estimates are calculated by the discounted sum of the differences between the actual loss decrease and the predicted loss decrease (see Alg. 1). The memory is updated whenever a new sample is encountered, and the full process is given in Alg. 2. The memory management strategy significantly outperforms reservoir sampling, especially with memory only training.

### 4.3 MEMORY ONLY TRAINING

ER uses joint training where half of the training batch is obtained from the online stream and the other half from memory. However, we argue that using the streamed samples directly will skew the training to favor the recent samples more. Thus, we propose to use samples only from the memory for training, without using the streamed samples. The memory works as a distribution stabilizer for streamed samples through the memory update process (see Sec. 4.2), and samples are used for training only with the memory. We observe that the memory only training improves the performance despite its simple nature. Note that this is different from Prabhu et al. (2020) as we train with the memory during the online stream but Prabhu et al. (2020) does not.

### 4.4 ADAPTIVE LEARNING RATE SCHEDULING

The exponential with reset scheduler resets the LR to the initial value when a new class is encountered. As the reset occurs regardless of the current LR value, it could result in a large change in LR value. We argue that such abrupt changes may harm the knowledge learned from previous samples.

Instead, we propose a new data-driven LR scheduling scheme that adaptively changes the LR in a data-driven manner based on how good the LR is for optimizing over the memory.

Specifically, from the current base LR $\bar{\eta}$ and step size $\gamma < 1$, we try both a high LR $\bar{\eta}/\gamma$ and a low LR $\bar{\eta} \cdot \gamma$ for training. For each LR, we keep a history of length $m$ that tracks the loss decrease for each LR. When both histories are full, we perform a Student's $t$-test with significance level $\alpha = 0.05$ to compare the LRs. If one LR is better, the base LR is set to the better LR, *i.e.*, $\frac{\bar{\eta}}{\gamma}$ or $\bar{\eta} \cdot \gamma$.

We depict this scheme in Alg. 3 in Sec. A.4. The adaptive LR is important for CL methods because the data distribution changes over time and adapting to the current training data is more useful.

With all the proposed components, we call our method **Continual Learning for i-Blurry** or (**CLIB**).

## 5 EXPERIMENTS

**Experimental Setup.** We use the CIFAR10, CIFAR100, TinyImageNet, and ImageNet datasets for empirical validations. We use the i-Blurry setup with $N = 50$ and $M = 10$ (i-Blurry-50-10) for our experiments unless otherwise stated. All results are averaged over 3 independent runs except ImageNet (Wu et al., 2019; Bang et al., 2021; Prabhu et al., 2020). For metrics, we use the average

| Methods | CIFAR10 | | CIFAR100 | | TinyImageNet | | ImageNet | |
|---|---|---|---|---|---|---|---|---|
| | $A_{\text{AUC}}$ | $A_{\text{avg}}$ | $A_{\text{AUC}}$ | $A_{\text{avg}}$ | $A_{\text{AUC}}$ | $A_{\text{avg}}$ | $A_{\text{AUC}}$ | $A_{\text{avg}}$ |
| Joint Training‡ (Soft Upper Bound) | 96.03 | | 79.89 | | 53.05 | | 69.26 | |
| EWC++ (Kirkpatrick et al., 2017) | 57.34±2.10 | 60.33±2.73 | 35.35±1.96 | 38.78±2.32 | 22.26±1.15 | 24.39±1.18 | 24.81 | 26.21 |
| BiC (Wu et al., 2019) | 58.38±0.54 | 61.49±0.68 | 33.51±3.04 | 37.61±3.00 | 22.80±0.94 | 24.90±1.07 | 27.41 | 28.38 |
| ER-MIR (Aljundi et al., 2019a) | 57.28±2.43 | 61.93±3.35 | 35.35±1.41 | 38.28±1.15 | 22.10±1.14 | 24.54±1.26 | 20.48 | 20.68 |
| GDumb (Prabhu et al., 2020) | 53.20±1.93 | 55.27±2.69 | 32.84±0.45 | 34.03±0.89 | 18.17±0.19 | 18.69±0.45 | 14.41 | 14.21 |
| RM† (Bang et al., 2021) | 23.00±1.43 | 61.52±3.69 | 8.63±0.19 | 33.27±1.59 | 5.74±0.30 | 17.04±0.77 | 6.22 | 28.30 |
| Baseline-ER (Sec. A.1) | 57.46±2.25 | 60.17±2.96 | 35.61±2.08 | 39.10±2.02 | 22.45±1.15 | 24.54±1.26 | 25.16 | 26.50 |
| **CLIB (Ours)** | **70.26±1.28** | **73.90±0.22** | **46.67±0.79** | **49.22±0.79** | **23.87±0.68** | **25.05±0.52** | **28.16** | **28.88** |

Table 1: Comparison of online CL methods on the i-Blurry setup for CIFAR10, CIFAR100, Tiny-ImageNet and ImageNet. 'Joint Training‡' shows the final accuracy of non-CL joint training as a soft upper bound where all relevant hyper-parameters were kept consistent with other compared CL methods. CLIB outperforms all other CL methods by large margins on both the $A_{\text{AUC}}$ and the $A_{\text{avg}}$.

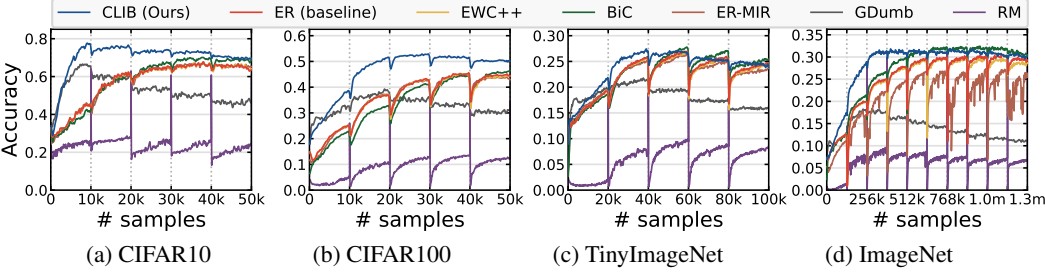

(a) CIFAR10    (b) CIFAR100    (c) TinyImageNet    (d) ImageNet

Figure 4: Accuracy-to-{number of samples} for various CL methods on CIFAR10, CIFAR100, TinyImageNet and ImageNet Our CLIB is consistent at maintaining high accuracy throughout inference while other CL methods are not as consistent.

accuracy ($A_{avg}$) and the proposed $A_{\text{AUC}}$ (see Sec. 3). Additional discussion with other metrics such as the forgetting measure ($F_{\text{last}}$) can be found in Sec. A.10.

**Implementation Details.** For all methods, we fix the batch size and the number of updates per streamed samples observed when possible. For CIFAR10, we use a batch size of 16 and 1 updates per streamed sample. When using ER, this translates to 8 updates using the same streamed batch, since each batch contains 8 streamed samples. For CIFAR100, we use a batch size of 16 and 3 updates per streamed sample. For TinyImageNet, we use a batch size of 32 and 3 updates per streamed sample. For ImageNet, we use a batch size of 256 and 1 update per every 4 streamed samples. We use ResNet-18 as the model for CIFAR10 and ResNet-34 for CIFAR100 and TinyImageNet. For all methods, we apply AutoAugment (Cubuk et al., 2019) and CutMix (Yun et al., 2019) following RM (Bang et al., 2021). For memory size, we use 500, 2000, 4000, 20000 for CIFAR10, CIFAR100, TinyImageNet, ImageNet, respectively. We use 5 tasks for CIFAR10, CIFAR100, and TinyImageNet and 10 tasks for ImageNet. We follow prior works (Bang et al., 2021; Prabhu et al., 2020) for choosing number of updates per sample, memory size, batch size, and number of tasks. Additional analysis on the sample memory size and number of tasks can be found in Sec. A.6. Adam optimizer with initial LR of 0.0003 is used. Exponential with reset LR schedule is applied for all methods except ours and GDumb, with $\gamma = 0.9999$ for CIFAR datasets and $\gamma = 0.99995$ for TinyImageNet and ImageNet. Ours use adaptive LR with $\gamma = 0.95, m = 10$ for all datasets, GDumb (Prabhu et al., 2020) and RM (Bang et al., 2021) follow original settings in their respective paper. All experiments were performed based on NAVER Smart Machine Learning (NSML) platform (Kim et al., 2018; Sung et al., 2017). All code and i-Blurry-$N$-$M$ splits (see Supp.) is at `https://github.com/naver-ai/i-Blurry`.

**Baselines.** We compare our method with both online CL methods and ones that can be extended to the online setting; EWC++ (Chaudhry et al., 2018b), BiC (Wu et al., 2019), GDumb (Prabhu et al., 2020), A-GEM (Chaudhry et al., 2019), MIR (Aljundi et al., 2019a) and RM (Bang et al., 2021). ·† indicates that the two-stage training scheme (Bang et al., 2021) is used. For details of the online versions of these methods, see Sec. A.5. Note that A-GEM performs particularly worse (also observed in (Prabhu et al., 2020; Mai et al., 2021)) as A-GEM was designed for the task-incremental setup and our setting is task-free. We discuss the comparisons to A-GEM in Sec. A.7.

## 5.1    RESULTS ON THE I-BLURRY SETUP

In all our experiments, we denote the best result for each of the metrics in **bold**.

| Varying $N$ | $N = 0$ (Blurry) | | $N = 50$ (i-Blurry) | | $N = 100$ (Disjoint) | |
|---|---|---|---|---|---|---|
| | $A_{\text{AUC}}$ | $A_{\text{avg}}$ | $A_{\text{AUC}}$ | $A_{\text{avg}}$ | $A_{\text{AUC}}$ | $A_{\text{avg}}$ |
| EWC++ | 53.24±0.56 | 57.04±0.94 | 57.34±2.10 | 60.33±2.73 | 77.64±1.81 | 77.80±1.93 |
| BiC | 51.51±0.12 | 54.88±0.30 | 58.38±0.54 | 61.49±0.68 | **78.78±1.52** | **80.12±2.20** |
| ER-MIR | 52.21±0.85 | 56.33±0.47 | 57.28±2.43 | 61.93±3.35 | 76.49±1.97 | 78.20±2.01 |
| GDumb | 45.86±0.80 | 46.37±2.09 | 53.20±1.93 | 55.27±2.69 | 65.27±1.54 | 66.74±2.39 |
| RM† | 22.54±1.11 | 54.07±0.70 | 23.00±1.43 | 61.52±3.69 | 33.17±3.71 | 66.67±2.38 |
| Baseline-ER | 53.28±0.57 | 57.13±1.01 | 57.46±2.25 | 60.17±2.96 | 77.82±2.06 | 77.47±2.69 |
| **CLIB (Ours)** | **68.87±0.83** | **72.79±0.96** | **70.26±1.28** | **73.90±0.22** | 78.58±2.09 | 77.96±3.28 |
| Varying $M$ | $M = 10$ | | $M = 30$ | | $M = 50$ | |
| | $A_{\text{AUC}}$ | $A_{\text{avg}}$ | $A_{\text{AUC}}$ | $A_{\text{avg}}$ | $A_{\text{AUC}}$ | $A_{\text{avg}}$ |
| EWC++ | 57.34±2.10 | 60.33±2.73 | 65.71±2.20 | 69.94±3.01 | 68.01±0.85 | 73.26±2.33 |
| BiC | 58.38±0.54 | 61.49±0.68 | 65.88±3.24 | 70.31±4.88 | 68.08±3.72 | 73.33±1.21 |
| ER-MIR | 57.28±2.43 | 61.93±3.35 | 65.99±2.28 | 70.47±3.41 | 68.13±0.65 | 73.33±1.21 |
| GDumb | 53.20±1.93 | 55.27±2.69 | 54.73±1.54 | 54.63±2.39 | 53.86±0.59 | 52.82±1.24 |
| RM† | 23.00±1.43 | 61.52±3.69 | 26.60±1.74 | 61.52±0.48 | 28.32±5.08 | 59.32±4.57 |
| Baseline-ER | 57.46±2.25 | 60.17±2.96 | 65.92±2.25 | 70.04±2.87 | 68.26±0.84 | 73.16±1.49 |
| **CLIB (Ours)** | **70.26±1.28** | **73.90±0.22** | **75.04±2.81** | **77.87±2.57** | **75.14±1.27** | **78.30±2.01** |

Table 2: Analysis on various values of $N$ (top) and $M$ (bottom) in the i-Blurry-$N$-$M$ setup using CIFAR10 dataset. For varying $N$, we use $M = 10$. For varying $M$, we use $N = 50$. Note that $N = 0$ corresponds to the blurry split and $N = 100$ corresponds to the disjoint split. For $N = 100$, CLIB outperforms or performs on par with other CL methods. For $N = 0$, the gap between CLIB and other methods widens. For $N = 50$, CLIB again outperforms all comparisons by large margins. For varying $M$, CLIB outperforms all comparisons excepting only the $A_{avg}$ when $M = 0$.

We first compare various online CL methods in the i-Blurry setup on CIFAR10, CIFAR100, TinyImageNet and ImageNet in Table 1. On CIFAR10, proposed CLIB outperforms all other CL methods by large margins; at least $+12.67\%$ in $A_{\text{AUC}}$ and $+12.85\%$ in $A_{avg}$. On CIFAR100, CLIB also outperforms all other methods by large margins; at least $+11.47\%$ in $A_{\text{AUC}}$ and $+10.18\%$ in $A_{avg}$. On TinyImageNet, all methods score very low. Nonetheless, CLIB outperforms other CL methods by at least $+2.71\%$ in $A_{\text{AUC}}$ and $+1.90\%$ in $A_{avg}$. Surprisingly on ImageNet, most methods perform slightly better than on TinyImagenet. We believe the reason is that the samples per class is the same with TinyImageNet (*e.g.*, 20) but ImageNet has higher resolution images, making learning from those images easier than TinyImageNet. On ImageNet, CLIB still outperforms other CL methods by at least $+2.60\%$ in $A_{\text{AUC}}$ and $+2.06\%$ in $A_{avg}$. Since, CLIB uses memory only training scheme, the distribution of the training samples are stabilized through the memory (see Sec. 4.3) which is helpful in the i-Blurry setup where the model encounters samples from more varied classes.

Note that Rainbow Memory (RM) (Bang et al., 2021) exhibits a very different trend than other compared methods. On TinyImageNet, it performs poorly. We conjecture that the delayed learning from the two-stage training is particularly detrimental in larger datasets with longer training duration and tasks such as TinyImageNet. On CIFAR10 and CIFAR100, RM performs reasonably well in $A_{avg}$ but poorly in $A_{\text{AUC}}$. This verifies that its two-stage training method delays most of the learning to the end of each task, resulting in a poor any-time inference performance measured by $A_{\text{AUC}}$. Note that $A_{avg}$ fails to capture this; on CIFAR10, the difference in $A_{avg}$ for CLIB and RM is $+13.27\%$ which is similar to other methods but the difference for $A_{\text{AUC}}$ is $+47.55\%$ which is noticeably larger than other methods. Similar trends can be found on other datasets as well.

We also show the accuracy-to-{# of samples} curve for the CL methods on CIFAR10, CIFAR100, TinyImageNet and ImageNet for comprehensive analysis throughout training in Fig 4. Interestingly, RM shows a surged accuracy only at the task transitions due to its two-stage training method and the accuracy is overall low. Additionally, GDumb shows a severe decreasing trend in accuracy as tasks progress. It is because the 'Dumb Learner' trains from scratch at every inference query leading to accuracy degradation. In contrast, CLIB not only outperforms other methods but also shows the most consistent accuracy at all times. More discussion of other methods are in Sec. A.9.

## 5.2 ANALYSIS ON DISJOINT CLASS PERCENTAGES ($N$) AND BLURRY LEVELS ($M$).

We further investigate the effect of different $N$ and $M$ values in the i-Blurry-N-M splits with various CL methods and summarize the results for varying values of the disjoint class percentages such as

| Methods | CIFAR10 | | CIFAR100 | |
|---|---|---|---|---|
| | $A_{\text{AUC}}$ | $A_{avg}$ | $A_{\text{AUC}}$ | $A_{avg}$ |
| CLIB | **70.26±1.28** | **73.90±0.22** | **46.67±0.79** | **49.22±0.79** |
| w/o Sample Importance Mem. (Sec.4.2) | 53.75±2.11 | 56.31±2.56 | 36.59±1.22 | 38.59±1.21 |
| w/o Memory-only training (Sec.4.3) | 67.06±1.51 | 71.65±1.87 | 44.63±0.22 | 48.66±0.39 |
| w/o Adaptive LR scheduling (Sec.4.4) | 69.70±1.34 | 73.06±1.35 | 45.01±0.22 | 48.97±0.12 |

Table 3: Ablations for proposed components of our method using CIFAR10 and CIFAR100 dataset. All proposed components improve the performance, with sample-wise importance based memory management providing the biggest gains. While adaptive LR scheduling provides small gains in CIFAR10, the gains increase in the more challenging CIFAR100.

$N = 0, 50, 100$ in Table 2 (top). For $N = 0$, CLIB outperforms other methods by at least $+16.59\%$ in $A_{\text{AUC}}$ and $+16.36\%$ in $A_{avg}$. For $N = 100$, the performance is similar for the majority of the methods, with CLIB being the best in $A_{\text{AUC}}$. For $N = 50$, CLIB outperforms all comparisons by at least $+12.55\%$ in $A_{\text{AUC}}$ and $+13.31\%$ in $A_{avg}$. Even though CLIB was designed with the i-Blurry setup in mind, it also outperforms other CL methods in conventional setups such as the $N = 100$ (disjoint) or the $N = 0$ (blurry) setups. It implies that CLIB is generally applicable to online CL setups and not restricted to just the i-Blurry setup. Meanwhile, except GDumb, all methods show similar performance on the $N = 100$ (disjoint) setup. The results imply that the i-Blurry setup differentiates CL methods better than the more traditionally used disjoint setup.

Following Bang et al. (2021), we additionally summarize the results for varying values of the blurry level such as $M = 10, 30, 50$ in Table 2 (bottom). We observe that CLIB again outperforms or performs on par with other CL methods on various blurry levels.

## 5.3 ABLATION STUDIES

We show the ablation studies (on CIFAR10/100) for each of the proposed components in Table 3.

**Sample-wise Importance Based Memory Management.** We replace the 'sample-wise importance memory management' module with the reservoir sampling. As shown in the table, the removal of our memory management strategy degrades the performance in both $A_{\text{AUC}}$ and $A_{avg}$ on both CIFAR10 and CIFAR100. As explained in Sec. 4.2, reservoir sampling removes samples at random, hence samples are discarded without considering if some samples are more important than others. Thus, using sample-wise importance to select which sample to discard greatly contributes to performance.

**Memory Only Training.** We replace the memory usage strategy from our memory only training with ER. Training with ER means that samples from both the online stream and the memory are used. Without the proposed memory only training scheme, the performance degrades across the board by fair margins. As the streamed samples are being used directly without the sample memory acting as a distribution regularizer (see Sec. 4.3), the CL model is more influenced by the recently observed samples, skewing the training and resulting in worse performance.

**Adaptive Learning Rate Scheduling.** We change the LR scheduling from adaptive LR to exponential with reset. The performance drop is larger in the more challenging CIFAR100 where more training iterations make the ablated model suffer more from a lack of good adaptive LR scheduling.

## 6 CONCLUSION

We question the practicality of existing continual learning setups for real-world application and propose a novel CL setup named i-Blurry. It is online, task-free, class-incremental, has blurry task boundaries, and is subject to any-time inference. Additionally, we propose a new metric to better evaluate the effectiveness of any-time inference. To address this realistic CL setup, we propose a method which uses per-sample memory management, memory only training, and adaptive LR scheduling, named Continual Learning for i-Blurry (CLIB). Our proposed CLIB consistently outperforms existing CL methods in multiple datasets and setting combinations by large margins.

ETHICS STATEMENT

All continual learning (CL) methods including the proposed one would adapt and extend the already trained AI model to recognize better with the streamed data. The CL methods will expedite the deployment of AI systems to help humans by its versatility of adapting to a new environment out of the factory or research labs. As all CL methods, however, would suffer from adversarial streamed data as well as data bias, which may cause ethnic, gender or biased gender issues, the proposed CLIB would not be an exception. Although the proposed CLIB has *no intention* to allow such problematic cases, the method may be exposed to such threats. Relentless efforts should be made to develop mechanisms to prevent such usage cases in order to make the continuously updating machine learning models safer and enjoyable to be used by humans.

REPRODUCIBILITY STATEMENT

We take the reproducibility of the research very seriously and release all codes, data splits and containers (*e.g.*, Docker) that include the general framework, learned models, and downstream tasks at https://github.com/naver-ai/i-Blurry.

ACKNOWLEDGEMENT

This work was partly supported by the National Research Foundation of Korea (NRF) grant funded by the Korea government (MSIT) (No.2022R1A2C4002300) and Institute for Information & communications Technology Promotion (IITP) grants funded by the Korea government (MSIT) (No.2020-0-01361-003 and 2019-0-01842, Artificial Intelligence Graduate School Program (Yonsei University, GIST), and No.2021-0-02068 Artificial Intelligence Innovation Hub)).

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

# A APPENDIX

## A.1 ADDITIONAL DISCUSSION FOR OUR BASELINE

In constructing our baseline, we used the reservoir sampling for memory management and ER for memory usage. We used reservoir sampling as it is widely used in the online and task-free setups with good performance. Note that in online CL, memory management policies that use the entire task's samples at once, such as herding selection (Rebuffi et al., 2017), mnemonics (Liu et al., 2020), and rainbow memory (Bang et al., 2021) are inapplicable.

For the memory usage, we use ER which draws half of the training batch from the stream and the other half from the memory, following a large number of online CL methods based on ER with good performance (Mai et al., 2021).

For the LR scheduling, we first note that other CL methods use either (1) exponential decay (Rebuffi et al., 2017; Kirkpatrick et al., 2017; Mirzadeh et al., 2020) or (2) constant LR. We do not use (1) as it is hyper-parameter sensitive; the decay rate that works for CIFAR10 decayed the LR too quickly for larger datasets such as CIFAR100. If the LR is decayed too fast, the LR becomes too small to learn about new classes that are introduced in the future. Thus, we use exponential LR scheduler with the modification that the LR is reset when a new class is observed. Comparing with the constant LR, we obtain slightly better performance for EWC++ and our baseline on CIFAR10, as shown in Table 9. Thus, we denote this LR schedule as the exponential with reset and use it in our baseline.

## A.2 PROOF OF THEOREM 1

We give the proof of Theorem 1 below. Our assumption is that when selecting memory $\mathcal{M}$ from a set of candidates $\mathcal{C}$, we should select $M$ so that optimizing on $M$ maximizes the loss decrease on $C$. In equations, optimal memory $\mathcal{M}^*$ is

$$\mathcal{M}^* = \underset{\mathcal{M} \subset \mathcal{C}, |\mathcal{M}| \leq m}{\arg\max} \sum_{(x',y') \in \mathcal{M}} \sum_{(x,y) \in \mathcal{C}} \mathbb{E}_\theta \left[ l(x,y;\theta) - l(x,y;\theta - \nabla_\theta l(x',y';\theta)) \right] \quad (4)$$

$$= \underset{\mathcal{M} \subset \mathcal{C}, |\mathcal{M}| \leq m}{\arg\max} \left[ \sum_{(x',y') \in \mathcal{C}} \sum_{(x,y) \in \mathcal{C}} \mathbb{E}_\theta \left[ l(x,y;\theta) - l(x,y;\theta - \nabla_\theta l(x',y';\theta)) \right] \right.$$

$$\left. - \sum_{(x',y') \in \mathcal{C} \backslash \mathcal{M}} \sum_{(x,y) \in \mathcal{C}} \mathbb{E}_\theta \left[ l(x,y;\theta) - l(x,y;\theta - \nabla_\theta l(x',y';\theta)) \right] \right] \quad (5)$$

$$= \underset{\mathcal{M} \subset \mathcal{C}, |\mathcal{M}| \leq m}{\arg\min} \sum_{(x',y') \in \mathcal{C} \backslash \mathcal{M}} \sum_{(x,y) \in \mathcal{C}} \mathbb{E}_\theta \left[ l(x,y;\theta) - l(x,y;\theta - \nabla_\theta l(x',y';\theta)) \right], \quad (6)$$

where $\theta$ is the model parameter, $l$ is the loss function, and $m$ is the memory size. Since we perform memory update after every streamed sample, the problem reduces to selecting one sample $(\bar{x}, \bar{y})$ to remove when the memory is full. Thus, $\mathcal{C} \backslash \mathcal{M} = \{(\bar{x}, \bar{y})\}$. The optimal removed sample $(\bar{x}^*, \bar{y}^*)$ would be

$$(\bar{x}^*, \bar{y}^*) = \underset{(\bar{x},\bar{y}) \in \mathcal{C}}{\arg\min} \mathbb{E}_\theta \left[ \sum_{(x,y) \in \mathcal{C}} l(x,y;\theta) - l(x,y;\theta - \nabla_\theta l(\bar{x},\bar{y};\theta)) \right]. \quad (7)$$

## A.3 DETAILS ON SAMPLE-WISE IMPORTANCE BASED MEMORY MANAGEMENT

We describe the details of our sample-wise importance based memory management here. We update $\mathcal{H}$, the estimate of sample-wise importance for episodic memory, after every model update. The details are in Alg. 1. The runtime complexity of updating the sample-wise importance is $\mathcal{O}(M)$ where $M$ is the fixed memory size. Because the memory size is kept constant regardless of the data size, the algorithm can scale to larger datasets. Note that for practical efficiency, Alg. 1 can be used after every $k$ model updates instead of every model update as described above. With the sample-wise importance scores, we update the memory everytime a new sample is encountered. The details are in Alg. 2.

---

**Algorithm 1** Update Sample-wise Importance

---

1: **Input** model $f_\theta$, memory $\mathcal{M}$, sample-wise importance $\mathcal{H}$, previous loss $l_{\text{prev}}$, indices used for training $\mathcal{I}$, update coefficient $\lambda$
2: $l_{\text{cur}} = \frac{1}{|\mathcal{M}|} \sum_{(x,y) \in \mathcal{M}} l(x, y; \theta)$        ▷ Obtain memory loss
3: $\Delta l = l_{\text{prev}} - l_{\text{cur}}$        ▷ Obtain memory loss decrease
4: $\Delta l_{\text{pred}} = \frac{1}{|\mathcal{I}|} \sum_{i \in \mathcal{I}} \mathcal{H}_i$        ▷ Memory loss decrease prediction using current $\mathcal{H}$
5: **for** $i \in \mathcal{I}$ **do**
6:      **Update** $\mathcal{H}_i \leftarrow \mathcal{H}_i + \lambda(\Delta l - \Delta l_{\text{pred}})$        ▷ Update $\mathcal{H}$ for samples used for training
7: **end for**
8: **Update** $l_{\text{prev}} \leftarrow l_{\text{cur}}$
9: **Output** $\mathcal{H}, l_{\text{prev}}$

---

**Algorithm 2** Sample-wise Importance Based Memory Update

---

1: **Input** model $f_\theta$, memory $\mathcal{M}$, memory size $m$, sample $(\hat{x}, \hat{y})$, per-sample criterion $\mathcal{H}$, previous loss $l_{\text{prev}}$
2: **if** $|\mathcal{M}| < m$ **then**        ▷ If the memory is not full
3:      **Update** $\mathcal{M} \leftarrow \mathcal{M} \cup \{(\hat{x}, \hat{y})\}$        ▷ Append the sample to the memory
4:      $\hat{i} = |\mathcal{M}|$
5: **else**        ▷ If the memory is already full
6:      $y_{\max} = \arg\max_y |\{(x_i, y_i)|(x_i, y_i) \in \mathcal{M} \cup \{(\hat{x}, \hat{y})\}, y_i = y\}|$    ▷ Find the most frequent label
7:      $\mathcal{I}_{y_{\max}} = \{i|(x_i, y_i) \in \mathcal{M}, y_i = y_{\max}\}$
8:      $\hat{i} = \arg\min_{i \in \mathcal{I}_{y_{\max}}} \mathcal{H}_i$        ▷ Find the sample with the lowest importance
9:      **Update** $l_{\text{prev}} \leftarrow \frac{m}{m-1} l_{\text{prev}} - \frac{1}{m-1} l(x_{\hat{i}}, y_{\hat{i}}; \theta)$
10:      **Update** $\mathcal{M}_{\hat{i}} \leftarrow (\hat{x}, \hat{y})$        ▷ Replace that sample with the new sample
11: **end if**
12: **Update** $l_{\text{prev}} \leftarrow \frac{|\mathcal{M}|-1}{|\mathcal{M}|} l_{\text{prev}} + \frac{1}{|\mathcal{M}|} l(\theta, \hat{x}, \hat{y})$
13: $\mathcal{I}_{\hat{y}} = \{i|(x_i, y_i) \in \mathcal{M}, y_i = \hat{y}, i \neq \hat{i}\}$
14: **Update** $\mathcal{H}_{\hat{i}} \leftarrow \frac{1}{|\mathcal{I}_{\hat{y}}|} \sum_{i \in \mathcal{I}_{\hat{y}}} \mathcal{H}_i$        ▷ Initialize the importance for the new sample
15: **Output** $\mathcal{M}, \mathcal{H}, l_{\text{prev}}$

---

### A.4 ADAPTIVE LEARNING RATE SCHEDULER

We describe the adaptive LR schedule from 4.4 in Alg. A.4. We fix the significance level to the commonly used $\alpha = 0.05$. Our adaptive LR scheduling decreases or increases the LR based on its current value. Thus, the rate in which the LR can change is bounded and sudden changes in LR do not happen.

### A.5 DETAILS ON THE ONLINE VERSIONS OF COMPARED CL METHODS

We implemented online versions of RM (Bang et al., 2021), EWC++ (Chaudhry et al., 2018b), BiC (Wu et al., 2019), GDumb (Prabhu et al., 2020), A-GEM (Chaudhry et al., 2019), GSS (Aljundi et al., 2019c) and MIR (Aljundi et al., 2019a) by incorporating ER and exponential decay with reset to the methods whenever possible. There is no specified memory management strategy for EWC++, and BiC uses herding selection from iCaRL(Rebuffi et al., 2017). However, herding selection is not possible in online since it requires whole task data for calculating class mean, so we attach reservoir memory to both methods instead. EWC++ does not require any other modification.

Additional modification should be applied to bias correction stage of BiC. BiC originally performs bias correction at end of each task, but since evaluation is also performed at the middle of task in our setup, we modified the method to perform bias correction whenever the model receives inference query.

In RM, their memory management strategy based on uncertainty is not applicable in an online setup, since it requires uncertainty rankings of the whole task samples. Thus, we replace their sampling strategy with balanced random sampling, while keeping their two-stage training scheme. Methods

---

**Algorithm 3** Adaptive Learning Rate Scheduler

---

1: **Input** current LR $\eta$, current base LR $\bar{\eta}$, loss before applying current LR $l_{\text{before}}$, current loss $l_{\text{cur}}$, LR performance history $\mathcal{H}^{(\text{high})}$ and $\mathcal{H}^{(\text{low})}$, LR step $\gamma < 1$, history length $m$, significance level $\alpha$
2: $l_{\text{diff}} = l_{\text{before}} - l_{\text{cur}}$ ▷ Obtain loss decrease
3: **Update** $l_{\text{before}} \leftarrow l_{\text{cur}}$
4: **if** $\eta > \bar{\eta}$ **then** ▷ If LR is higher than base LR
5:     **Update** $\mathcal{H}^{(\text{high})} \leftarrow \mathcal{H}^{(\text{high})} \cup \{l_{\text{diff}}\}$ ▷ Append loss decrease in high LR history
6:     **if** $|\mathcal{H}^{(\text{high})}| > m$ **then**
7:         **Update** $\mathcal{H}^{(\text{high})} \leftarrow \mathcal{H}^{(\text{high})} \setminus \{\mathcal{H}_1^{(\text{high})}\}$
8:     **end if**
9: **else** ▷ If LR is lower than base LR
10:     **Update** $\mathcal{H}^{(\text{low})} \leftarrow \mathcal{H}^{(\text{low})} \cup \{l_{\text{diff}}\}$ ▷ Append loss decrease in low LR history
11:     **if** $|\mathcal{H}^{(\text{low})}| > m$ **then**
12:         **Update** $\mathcal{H}^{(\text{low})} \leftarrow \mathcal{H}^{(\text{low})} \setminus \{\mathcal{H}_1^{(\text{low})}\}$
13:     **end if**
14: **end if**
15: **if** $|\mathcal{H}^{(\text{high})}| = m$ **and** $|\mathcal{H}^{(\text{low})}| = m$ **then** ▷ If both histories are full
16:     $p = \text{OneSidedStudentsTTest}\left(\mathcal{H}^{(\text{low})}, \mathcal{H}^{(\text{high})}\right)$
                           ▷ Perform one-sided Student's $t$-test with alternative hypothesis $\mu_{\text{low}} > \mu_{\text{high}}$
17:     **if** $p < \alpha$ **then** ▷ If pvalue is significantly low
18:         **Update** $\bar{\eta} \leftarrow \gamma^2 \cdot \bar{\eta}$ ▷ Decrease base LR
19:         **Update** $\mathcal{H}^{(\text{low})}, \mathcal{H}^{(\text{high})} = \emptyset, \emptyset$ ▷ Reset histories
20:     **else if** $p > 1 - \alpha$ **then** ▷ If pvalue is significantly high
21:         **Update** $\bar{\eta} \leftarrow \frac{1}{\gamma^2} \cdot \bar{\eta}$ ▷ Increase base LR
22:         **Update** $\mathcal{H}^{(\text{low})}, \mathcal{H}^{(\text{high})} = \emptyset, \emptyset$
23:     **end if**
24: **end if**
25: **if** $\eta > \bar{\eta}$ **then** ▷ Alternately apply high and low LR (note that $\gamma < 1$)
26:     **Update** $\eta = \gamma \cdot \bar{\eta}$
27: **else**
28:     **Update** $\eta = \frac{1}{\gamma} \cdot \bar{\eta}$
29: **end if**
30: **Output** $\eta, \bar{\eta}, l_{\text{before}}, \mathcal{H}^{(\text{low})}, \mathcal{H}^{(\text{high})}$

---

that were converted from offline to online, namely EWC++, BiC, and RM, may have suffered some performance drop due to deviation from their original methods.

## A.6   Analysis on Sample Memory Size and Number of Tasks

We conduct analysis over various sample memory sizes ($K$) and summarize results in Table 4. We observe that CLIB outperforms other CL methods in both $A_{\text{AUC}}$ and $A_{avg}$ no matter the memory size. It is interesting to note that CLIB with a memory size of $K = 200$ outperforms other CL methods with a memory size of $K = 1000$ in the $A_{\text{AUC}}$ and performs on par in $A_{avg}$. Thus, CLIB is the only method using memory only training scheme but is the least sensitive to memory size. It implies that our memory management policy is the most effective, which shows the superiority of our per-sample memory management method.

We additionally conduct analysis on the number of tasks used for the CIFAR100 dataset and summarize the results in Table 5. Note that we use CIFAR100 as CIFAR10 has too few classes to divide into longer task sequences. We observe that CLIB outperforms other CL methods in both $A_{\text{AUC}}$ and $A_{avg}$ even when the number of tasks increase from 5 to 10 or 25. Moreover, CLIB shows only minor performance drops with increasing number of tasks, indicating that CLIB is capable of achieving high performance in the long-run.

Interestingly, we note that BiC shows a slight performance increase when the number of tasks is increased to 10 or 25. We believe this is because BiC uses separate bias parameters for

| Methods | K=200 | | K=500 | | K=1000 | |
|---|---|---|---|---|---|---|
| | $A_{\text{AUC}}$ | $A_{\text{avg}}$ | $A_{\text{AUC}}$ | $A_{\text{avg}}$ | $A_{\text{AUC}}$ | $A_{\text{avg}}$ |
| EWC++ | 52.06±2.24 | 54.09±3.57 | 57.34±2.10 | 60.33±2.73 | 60.93±1.02 | 65.86±2.05 |
| BiC | 53.00±1.03 | 54.36±1.64 | 58.38±0.54 | 61.49±0.68 | 61.52±2.24 | 64.82±1.15 |
| ER-MIR | 51.63±2.43 | 54.40±3.50 | 57.28±2.43 | 61.93±3.35 | 61.18±1.08 | 66.05±2.29 |
| GDumb | 42.54±2.01 | 43.99±2.28 | 53.20±1.93 | 55.27±2.69 | 66.55±1.10 | 69.21±1.29 |
| RM$^\dagger$ | 21.24±1.35 | 46.79±3.78 | 23.00±1.43 | 61.52±3.69 | 26.13±1.61 | 72.29±2.17 |
| Baseline-ER | 52.11±2.32 | 54.34±3.34 | 57.46±2.25 | 60.17±2.96 | 61.18±1.08 | 66.05±2.29 |
| **CLIB (Ours)** | **64.67±1.86** | **66.06±1.78** | **70.26±1.28** | **73.90±0.22** | **73.00±1.30** | **77.87±1.15** |

Table 4: Analysis on various sample memory sizes ($K$) using CIFAR10. The i-Blurry-50-10 splits are used. The results are averaged over 3 runs. CLIB outperforms all other CL methods by large margins for all the memory sizes. CLIB uses the given memory budget most effectively, showing the superiority of our per-sample memory management method.

| Methods | 5 Tasks | | 10 Tasks | | 25 Tasks | |
|---|---|---|---|---|---|---|
| | $A_{\text{AUC}}$ | $A_{\text{avg}}$ | $A_{\text{AUC}}$ | $A_{\text{avg}}$ | $A_{\text{AUC}}$ | $A_{\text{avg}}$ |
| EWC++ | 35.35±1.96 | 38.78±2.32 | 32.25±1.56 | 34.85±1.71 | 27.08±1.56 | 29.46±1.65 |
| BiC | 33.51±3.04 | 37.61±3.00 | 35.29±1.07 | 37.40±1.04 | 34.68±1.16 | 35.55±1.38 |
| ER-MIR | 35.35±1.41 | 38.28±1.15 | 33.65±1.51 | 35.29±1.56 | 28.16±1.29 | 30.00±1.17 |
| GDumb | 32.84±0.45 | 34.03±0.89 | 31.22±0.51 | 32.35±1.73 | 30.79±0.90 | 31.87±1.08 |
| RM$^\dagger$ | 8.63±0.19 | 33.27±1.59 | 6.56±0.43 | 34.93±3.94 | 3.66±0.10 | 36.85±1.17 |
| Baseline-ER | 35.61±2.08 | 39.10±2.02 | 32.46±1.35 | 34.76±1.34 | 28.35±1.56 | 30.45±1.43 |
| **CLIB (Ours)** | **46.67±0.79** | **49.22±0.79** | **44.61±1.16** | **46.15±1.07** | **43.09±1.04** | **43.64±1.08** |

Table 5: Analysis on the number of tasks on CIFAR100. The i-Blurry-50-10 splits are used. The results are averaged over 3 runs. CLIB outperforms all other CL methods by large margins, even when the task sequences becomes longer. Additionally, CLIB does not suffer severe performance drops as the number of tasks increase, indicating that CLIB is well-suited for long-run CL problem setups as well.

each task, longer task sequences allows more fine-grained bias correction and may actually be favorable for BiC. We also point out that the absolute performance of BiC is still lacking. Thus, it may imply that the bias correction of BiC contributes to strong stability in the stability-plasticity trade-off (Chaudhry et al., 2018a), which help maintain performance for longer task sequences where forgetting would be more severe. However, as stability is enforced strongly, BiC might lack plasticity to sufficiently learn from new data, resulting in low performance overall.

We also note that EWC++ and Baseline-ER have the same $A_{AUC}$ and $A_{avg}$ values up to the second decimal places in the 10 tasks setting. This may imply that EWC++ is seldom beneficial over Baseline-ER as the added episodic memory negates the need for the regularization in EWC++.

## A.7 Additional Comparisons with A-GEM

We present additional comparisons to A-GEM. Note that as A-GEM was designed for the task-incremental setting, it performs very poorly in our i-Blurry setup which is task-free. Notably, it achieves only 4.62 $A_{AUC}$ and 6.94 $A_{avg}$ on CIFAR100, but other works (Prabhu et al., 2020; Mai et al., 2021) have also reported very poor performance for A-GEM in their studies as well.

## A.8 Comparisons to Other Memory Management CL Methods

We present additional comparison to CL methods that use a different memory management strategy in Table 7. GSS (Aljundi et al., 2019c) is added as an additional comparison while Baseline-ER is used to represent the reservoir sampling. CLIB outperforms both methods by large margins in both $A_{AUC}$ and $A_{avg}$, implying that the sample-wise importance memory management method is better than reservoir or GSS-greedy.

| Methods | CIFAR10 | | CIFAR100 | |
|---|---|---|---|---|
| | $A_\text{AUC}$ | $A_\text{avg}$ | $A_\text{AUC}$ | $A_\text{avg}$ |
| EWC++ | 57.34±2.10 | 60.33±2.73 | 35.35±1.96 | 38.78±2.32 |
| BiC | 58.38±0.54 | 61.49±0.68 | 33.51±3.04 | 37.61±3.00 |
| ER-MIR | 57.28±2.43 | 61.93±3.35 | 35.35±1.41 | 38.28±1.15 |
| A-GEM | 39.29±2.88 | 44.85±4.70 | 4.62±0.23 | 6.94±0.51 |
| GDumb | 53.20±1.93 | 55.27±2.69 | 32.84±0.45 | 34.03±0.89 |
| RM$^\dagger$ | 23.00±1.43 | 61.52±3.69 | 8.63±0.19 | 33.27±1.59 |
| Baseline-ER | 57.46±2.25 | 60.17±2.96 | 35.61±2.08 | 39.10±2.02 |
| **CLIB (Ours)** | **70.26±1.28** | **73.90±0.22** | **46.67±0.79** | **49.22±0.79** |

Table 6: Additional comparisons to A-GEM with various online CL methods on the i-Blurry setup for CIFAR10 and CIFAR100 are shown. The i-Blurry-50-10 splits are used for all the datasets and the results are averaged over 3 runs. A-GEM performs very poorly, especially on CIFAR100 as it was designed for the task-incremental setting whereas i-Blurry setup is task-free. CLIB outperforms all other CL methods by large margins on both the $A_\text{AUC}$ and the $A_{avg}$.

| Methods | Mem. Management | CIFAR10 | | CIFAR100 | |
|---|---|---|---|---|---|
| | | $A_\text{AUC}$ | $A_\text{avg}$ | $A_\text{AUC}$ | $A_\text{avg}$ |
| GSS | GSS-Greedy | 55.51±3.33 | 59.27±4.36 | 30.09±1.38 | 35.06±1.43 |
| Baseline-ER | Reservoir | 57.46±2.25 | 60.17±2.96 | 35.61±2.08 | 39.10±2.02 |
| **CLIB (Ours)** | Sample-wise Importance | **70.26±1.28** | **73.90±0.22** | **46.67±0.79** | **49.22±0.79** |

Table 7: Comparisons to other CL methods with different memory management strategies in the i-Blurry setup for CIFAR10 and CIFAR100 are shown. The i-Blurry-50-10 splits are used for all the datasets and the results are averaged over 3 runs. CLIB outperforms all other CL methods by large margins on both the $A_\text{AUC}$ and the $A_{avg}$ implying that the sample-wise importance memory management method is effective.

## A.9 ADDITIONAL DISCUSSION OF OTHER METHODS

We present additional discussion on compared methods in Table 1.

### A.9.1 BASELINE-ER

ER has been reported to be a simple yet strong method for online CL. (Prabhu et al., 2020; Mai et al., 2021) We also observed strong performance for Baseline-ER, as other methods such as EWC++, BiC, and MIR only show marginal to no improvements. However, by using different memory management and memory usage strategy, CLIB was able to outperform Baseline-ER by large margins

### A.9.2 EWC++

EWC is a regularization-based method, which regularizes parameters based on the parameter's importance measured by the accumulated Fisher Information. Note that EWC calculates importance of parameters to regularize them, while CLIB calculates importance of samples to manage episodic memory. We use an online version of EWC, called EWC++, proposed in (Chaudhry et al., 2018b). EWC++ shows almost no improvement over Baseline-ER, possibly because it was developed as a method to prevent forgetting without using episodic memory. Since episodic memory with ER alleviates most of the forgetting, EWC++ have reduced effect in preventing forgetting, while its side-effect of reducing intransigence still exists (Chaudhry et al., 2018b).

### A.9.3    BIC

BiC uses episodic memory and a distillation loss for training and has an additional bias-correction step to correct the model's bias towards current task's classes. BiC shows strong performance on the disjoint task split. However, it performs poorly for the blurry and the i-Blurry splits, possibly because the bias correction using only two parameters for previous and current tasks cannot correctly capture the bias in the presence of blurry classes.

### A.9.4    ER-MIR

Note that ER-MIR performs surprisingly similar to the Baseline-ER for CIFAR10, possibly because they are both based on ER. However, ER-MIR does not scale well to larger datasets potentially because its memory usage scheme is not effective for larger datasets.

To extensively highlight the differences between MIR and CLIB, we first have to point out that sample memory methods can be discussed in two aspects: memory management and memory usage. MIR uses reservoir sampling for memory management and an improved version of experience replay (ER) for memory usage. MIR improves the memory usage by selecting samples from memory that would suffer the largest loss increase if streamed data were used to train the model (called 'maximally interfered retrieval' or MIR).

MIR can be viewed as using some form of 'importance weighting', and has some similarities with our method in the sense that they both utilize the change in loss to determine which samples are used for training. However, these two are different as MIR uses importance scores for selecting samples from a constructed memory whereas CLIB uses it for constructing the memory. Also, MIR assigns high importance scores to samples that show large increases in individual loss when current streamed data are used for training. In contrast, CLIB assigns high importance scores to samples that cause large decreases in memory's total loss when that sample is used for training. Lastly, MIR's importance score is temporary, as new importance scores are calculated every iteration, while CLIB accumulates the measured importance scores over time to calculate the overall importance scores.

Our method performs better than MIR possibly because sample-wise importance memory considers each samples' importance whereas reservoir sampling does not and memory-only training is not skewed by recently seen samples whereas joint training is. Note that since we are selecting random samples in the memory usage step, a possible future direction would be improving the memory usage as done in MIR.

### A.9.5    GDUMB

GDumb utilizes a greedy sampler, which stores samples greedily, and a dumb learner, which trains a new model from scratch using only the memory whenever an inference query is made. While we also train only using the memory, we have a completely different memory management method that updates both the model and the memory during training. Because GDumb trains from scratch, its performance degrades in the later phases as it cannot accumulate past knowledge. Not only that, because GDumb has to train from scratch for every inference query, it takes more computation than other methods when there are many inference queries for any-time inference.

### A.9.6    RAINBOW MEMORY

Rainbow memory uses uncertainty based memory management with a two-stage training method that prolongs most of the training until the end of a task. As such, RM performs very poorly on our $A_{AUC}$ metric as it measures any-time inference performance. Nevertheless, RM performs strongly with respect to the $A_{avg}$ metric which shows the usefulness of our proposed $A_{AUC}$.

| Methods | CIFAR10 | | | CIFAR100 | | | TinyImageNet | | | ImageNet | | |
|---|---|---|---|---|---|---|---|---|---|---|---|---|
| | $A_{\mathrm{AUC}}$ | $A_{\mathrm{avg}}$ | $F_{\mathrm{last}}$ | $A_{\mathrm{AUC}}$ | $A_{\mathrm{avg}}$ | $F_{\mathrm{last}}$ | $A_{\mathrm{AUC}}$ | $A_{\mathrm{avg}}$ | $F_{\mathrm{last}}$ | $A_{\mathrm{AUC}}$ | $A_{\mathrm{avg}}$ | $F_{\mathrm{last}}$ |
| EWC++ | 57.34±2.10 | 60.33±2.73 | 28.84±2.35 | 35.35±1.96 | 38.78±2.32 | 32.11±5.28 | 22.26±1.15 | 24.39±1.18 | 31.65±1.50 | 24.81 | 26.21 | 44.08 |
| BiC | 58.38±0.54 | 61.49±0.68 | 22.50±4.38 | 33.51±3.04 | 37.61±3.00 | 23.10±2.87 | 22.80±0.94 | 24.90±1.07 | 27.01±0.83 | 27.41 | 28.38 | 36.12 |
| ER-MIR | 57.28±2.43 | 61.93±3.35 | 27.10±2.17 | 35.35±1.41 | 38.28±1.15 | 31.76±4.14 | 22.10±1.14 | 24.54±1.26 | 34.57±1.39 | 20.48 | 20.68 | 66.96 |
| GDumb | 53.20±1.93 | 55.27±2.69 | 18.80±3.40 | 32.84±0.45 | 34.03±0.89 | 22.44±3.52 | 18.17±0.19 | 18.69±0.45 | 17.14±1.13 | 14.41 | 14.21 | 31.18 |
| RM† | 23.00±1.43 | 61.52±3.69 | **7.01±2.74** | 8.63±0.19 | 33.27±1.59 | **7.76±1.21** | 5.74±0.3 | 17.04±0.77 | **9.40±0.40** | 6.22 | 28.3 | **10.71** |
| Baseline-ER | 57.46±2.25 | 60.17±2.96 | 30.20±2.96 | 35.61±2.08 | 39.10±2.02 | 31.31±1.96 | 22.45±1.15 | 24.54±1.26 | 33.02±1.10 | 25.16 | 26.50 | 44.22 |
| CLIB (Ours) | **70.26±1.28** | **73.90±0.22** | 23.08±2.68 | **46.67±0.79** | **49.22±0.79** | 23.93±0.66 | **23.87±0.68** | **25.05±0.52** | 23.34±1.96 | **28.16** | **28.88** | 38.66 |

Table 8: Additional comparisons including the $F_{\mathrm{last}}$ measure of various online CL methods on the i-Blurry setup for CIFAR10, CIFAR100, TinyImageNet and ImageNet are shown. The i-Blurry-50-10 splits are used for all the datasets and the results are averaged over 3 runs except ImageNet. Ours outperforms all other CL methods by large margins on both the $A_{\mathrm{AUC}}$ and the $A_{avg}$. While CLIB does not explicitly handle forgetting, it shows similar $F_{\mathrm{last}}$ with most methods. The best result for each of the metrics is shown in **bold**.

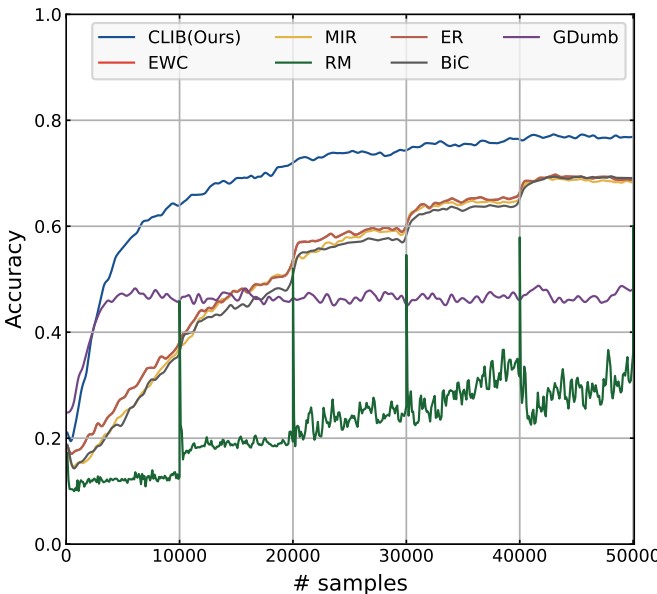

Figure 5: Accuracy-to-# of samples curve for the blurry setup (no new classes are encountered after the first task). We can see that there is no performance drop for all methods because no new classes are encountered after the first task.

## A.10 ADDITIONAL RESULTS WITH THE $F_{\mathrm{LAST}}$ MEASURE

We report the result of the forgetting measure (Chaudhry et al., 2018b) here. As in the i-Blurry setup it is not clear which classes belong to each task, we calculate the forgetting class-wise. Note that while forgetting is a useful metric for analyzing stability-plasticity of the method, lower forgetting does not necessarily mean that a CL method is better. For example, if a method do not train with the new task at all, its forgetting will be 0.

Also, we do not propose a new forgetting measure for anytime inference. It is because forgetting is measured with the best accuracy of each class, and best accuracy usually occur at the end of each task. Thus, measuring the best accuracy among all inferences would not be much different from best accuracy among the inferences at the end of each task.

Surprisingly, CLIB shows similar $F_{\mathrm{last}}$ values to most other methods despite not explicitly designed to handle catastrophic forgetting. We believe that the episodic memory used in our memory only training reduces catastrophic forgetting of our method to be similar to other methods.

| Methods | LR Schedule | $A_{\mathrm{AUC}}$ | $A_{\mathrm{avg}}$ |
|---|---|---|---|
| Baseline-ER | Constant | 56.73±1.73 | 58.18±3.41 |
| | Exp w/ Reset | 57.46±2.25 | 60.17±2.96 |
| EWC++ | Constant | 56.54±1.57 | 57.92±3.26 |
| | Exp w/ Reset | 57.34±2.10 | 60.33±2.73 |

Table 9: Comparison between exponential with reset schedule and constant LR on CIFAR10 are shown. It shows that our baseline LR schedule, exponential with reset, is reasonable. It shows better performance than constant LR, especially in $A_{\mathrm{avg}}$ metric.

Interestingly, RM shows the best $F_{\mathrm{last}}$ results on all datasets. We believe it is because of RM's two-stage training scheme, which greatly reduces the effect of the streamed data during training. This prevents dominant classes in the task from achieving very high accuracy. Since forgetting is calculated as the difference between the best accuracy and the last accuracy for each class, the forgetting is reduced as the best accuracy itself has been lowered.

Note that GDumb is even less affected by streamed data as it does not train with streamed data at all. However, GDumb does not accumulate learned knowledge, leading to decreasing accuracy as the number of samples per class in the memory decreases. Thus, while GDumb scores relatively good in forgetting considering the decreasing accuracy trend, it ultimately scores worse in forgetting than RM.

### A.11 ADDITIONAL DISCUSSION ON POTENTIAL PERFORMANCE DROP AT LATER PHASES

We discuss the potential cause for the 'performance drop' of CLIB in the i-Blurry setup as is shown in Fig. 4. We argue that the 'performance drop' is attributed to the number of encountered classes increasing in i-Blurry setups at later phases. To see what would happen when the number of encountered classes do not increase at later phases, we show the accuracy-to-# of samples curve for the blurry setup results in Table 2 for $N = 0$ in Fig. 5, where no new classes are encountered after the first task. There is no apparent performance drop at later phases for all methods, which supports our argument that the drop is due to more and more classes being encountered at later phases.

### A.12 PERFORMANCE OF EXPONENTIAL WITH RESET LR SCHEDULE

We show brief results for the LR schedule used in our baseline in Table 9. We compare the constant LR with the exponential with reset used in our baseline. The exponential with reset is better than the constant LR, which is why we used it in our baseline.

