# OpenReview forum: "Online Continual Learning on Class Incremental Blurry Task Configuration with Anytime Inference"
_ICLR.cc/2022/Conference — ICLR 2022 Poster_

### Official Review · Reviewer_Vfw2 · 2021-10-28

**Correctness:** 4
**Technical Novelty And Significance:** 3
**Empirical Novelty And Significance:** 3
**Recommendation:** 6
**Confidence:** 3

**Main Review:**

The main contribution of the paper, I believe, is the memory update method, which (as stated above) updates the memory when new samples arrive in a way that preserves the most "useful" samples (wrt to their effect on the loss) while promoting class balance. The authors also train exclusively on samples drawn from the memory, rather than on batches split between memory and newly arrived samples, which seems to be advantageous. And the authors use a learning-rate scheduling method that also gives some advantages. The memory management method is novel, as far as I'm aware, and the overall method performs very well. Evaluation is carried out in the class incremental, online, task-free setting, which is a good decision in my opinion. So many other methods fall short under these conditions.  So I think the paper deserves to be accepted.

The authors make a big deal out of their "blurry" sampling method for evaluation. I think it's a good approach, but isn't really a big contribution in its own right.

Presumably, following memory update, the batch that is trained on is sampled uniformly from the memory. Is that right?

Space permitting, it would be nice to see a more detailed comparison with related experience replay methods. In particular, Aljundi et al's MIR method also uses a kind of importance weighting. It would be good to see the differences spelled out, and to see some sort of explanation for why the present method performs better.

**Summary Of The Paper:**

The paper presents an experience replay method for continual learning (CL) whose main innovation is a memory-management algorithm that updates the memory when new samples arrive in a way that preserves the most "useful" samples (wrt to their effect on the loss) while promoting class balance. The method is evaluated on the trickiest CL setting - class incremental, online, and task-free - where it outperforms baselines on CIFAR-10 and -100 datasets.

**Summary Of The Review:**

A nice paper describing an effective and somewhat novel replay-based CL method, evauated in a demanding setting.

---

> ### Author Response · Authors · 2021-11-13
> **Answer to the questions of Reviewer Vfw2**
>
> Thank you for the encouraging remarks on challenging set-ups and evaluation choices. We address your comments as follows.
>
> > **Space permitting, it would be nice to see a more detailed comparison with related experience replay methods.** In particular, Aljundi et al's MIR method also uses a kind of importance weighting. It would be good to see the differences spelled out, ...
>
> $\to$ In the table below, we summarize the differences of the two methods in the two aspects of sample memory based CL methods; memory management and memory usage.
>
> |    Method   	|    Memory Management   	|         Memory Usage         	|
> |:-----------:	|:----------------------:	|:----------------------------:	|
> |    ER-MIR   	|        Reservoir       	|           ER + MIR           	|
> | CLIB (Ours) 	| Sample-wise Importance 	| Memory Only + Uniform Random 	|
> | | | |
>
> MIR uses reservoir sampling for memory management and an improved version of experience replay (ER) for memory usage. MIR improves the memory usage by selecting samples *from* memory that would suffer the largest loss increase if streamed data were used to train the model (called ‘maximally interfered retrieval’ or MIR).
>
> In contrast, our method uses ‘sample-wise importance sampling’ to *construct* the memory. Our memory management scheme keeps samples that cause the largest loss decrease for samples in the memory. Our memory usage scheme uses memory-only training with uniformly random sample selection. We agree with the reviewer that MIR can be viewed as using a form of ‘importance weighting’, and shares some similarities with our method in the sense that they both utilize the change in loss to determine which samples are used for training. However, the two are different by the following reasons:
> - MIR assigns high importance scores to samples that show large increases in individual loss when current streamed data are used for training.
>    - In contrast, our method assigns high importance scores to samples that cause large decreases in loss computed with all samples in memory when they are used for training.
> - MIR’s importance score is temporary as new importances scores are calculated at every iteration.
>    - In contrast, our method accumulates the measured importance scores over time to calculate the overall importance scores.
>
> We added the discussion to A.9.4 in revision due to space limitations.
>
> > **Space permitting, it would be nice to see a more detailed comparison with related experience replay methods.** In particular, Aljundi et al's MIR method also uses a kind of importance weighting. It would be good ... to see some sort of explanation for why the present method performs better.
>
> $\to$ One possible reason that our method outperforms MIR is that our sample-wise importance memory considers each samples’ importance when discarding samples from the memory whereas reservoir sampling used in MIR removes the samples randomly from the memory (*i.e.*, treating all samples equally). Also, memory-only training provides a momentum for abrupt data distribution changes as the streamed samples are passed through the memory that is updated incrementally (*i.e.*, one sample is updated at a time) whereas MIR uses the streamed samples directly (*i.e.*, without buffering in a memory) for training and hence when the data distribution drastically changes, training could become unstable.
>
> Note that a possible future work for our method would be improving the memory usage by using MIR or other memory usage schemes instead of uniform random sampling.
>
> We added the discussion to A.9.4 in revision due to space limitations.
>
>
> > **The authors make a big deal out of their "blurry" sampling method for evaluation.** I think it's a good approach, but isn't really a big contribution in its own right.
>
> $\to$ As the CL is motivated for a realistic learning scenario, increasing attention is drawn to CL methods which address the more realistic blurry setup, such as GSS (Aljundi et al. 2019) and RM (Bang et al., 2021). Our proposed i-Blurry setup could be appreciated as it is more realistic than the blurry setup.
>
> > **Presumably, following memory update, the batch that is trained on is sampled uniformly from the memory.** Is that right?
>
> $\to$ Yes, you are correct.

---

> > ### Comment · Reviewer_Vfw2 · 2021-11-15
> > **Response to response**
> >
> > Many thanks to the authors for these clarifications.

---

> > > ### Author Response · Authors · 2021-11-23
> > > **Thank you to Reviewer Vfw2**
> > >
> > > Thank you for your positive feedback and active participation in the discussion. Your participation greatly helps in clarifying our CL setup and contribution. We refer the reviewer to our reply in the above discussion. Please let us know if you have more comments.

---

### Official Review · Reviewer_6BAu · 2021-11-01

**Correctness:** 3
**Technical Novelty And Significance:** 3
**Empirical Novelty And Significance:** 3
**Recommendation:** 8
**Confidence:** 3

**Main Review:**

### Strengths:

1. The proposed method works well on a various instantiations of the i-Blurry-N-M setup. The proposed algorithms is better in the blurry (N=0) setting and performs comparably well in disjoint setting (N=100) as well.

2. The ablation study highlights the main component in the proposed approach, the memory management scheme that relies on sample importance, that results in the highest benefits.

3. Lots of supporting experiments are provided in the appendix that further corroborate most claims.

4. Good organization of related work and great figures.

### Weaknesses:

1. I am unsure if the proposed approach of obtaining sample importance is entirely novel. A discussion regarding this is necessary as it is likely that similar ideas present (beyond CL for instance) in literature.

2. A natural language description of Theorem 1 is missing.

3. I felt that more insights into the calculation of importance are required in the main text. A note about the efficiency/runtime needs to be specified.

**Personally**, I think dedicating a portion of the main-text (Section 4) in discussing these would probably be more beneficial than describing memory only training and adaptive lr, as they seemingly play only a minor role (as indicated by the ablations).

4. No information as to why TinyImagenet is challenging for all methods is provided. There is also no justification provided for choosing TinyImagenet over Imagenet (RM, BiC and GDumb all show their results on Imagenet).

5. Since the i-Blurry setting is an **online** one, the choice of hyperparameters are likely to affect the results greatly. As such hyperparameter choices (updates per sample, memory size and batchsize) need to justified. Specifically, were the hyperparameters chosen consistent with previous work?

6. For a fixed $M$, is it reasonable to expect that performance of all algorithms would increase (in general) with increasing N? This can be used to justify choosing $N$ to be from $0, 50, 100$ (would also be helpful reference for future works).

**Suggestion** (not a weakness): The paper can also report the performance on some (soft) upper bounds. For instance, what if the task was not restricted to be online. What is the performance when the entire training data is available at once (non CL scenario)?


##### There are some typos:

* Section 4.2 (page 5): Arguing that considering ... lead to training efficacy **decrease**.

* Section 4.1 (page 5): For the LR scheduling, other **CL methods use either (1) exponential decay** or (2) constant LR ...

* Last line on page 4 is missing a period (full-stop)

* Also, Table 3 $A_{avg}$ on CIFAR-100 has the same numbers for the last two rows, which might be a typo.

**Summary Of The Paper:**

The paper proposes a more realistic setting (called *i-Blurry*) for continual learning (CL) that generalizes the *blurry* and *disjoint*  settings proposed in prior work. The disjoint setting assumes that there is no class that appears in multiple tasks and the blurry setting assumes that no new classes are seen after the first task. In the *i-Blurry* setting, one can have overlapping classes across tasks as well as new classes appearing in each task. This setting is also **online** and thus the paper is interested in continuous model evaluation i.e., any-time inference too. It proposes a metric which calculates the area under the accuracy curve during training for the same.

For the new configuration, the paper proposes a strong baseline and a new algorithm called CLIB. CLIB is a memory-based CL method that refines its memory by throwing out *least important* samples and updating it with more important ones. The sample importance is calculated as the expected decrease in training loss when the sample is used for training. CLIB is also equipped with a data-driven adaptive learning rate scheduling scheme which provides some additional performance benefits.

The results showcase that CLIB is able to outperform other online CL methods and the baseline by large margins for various instantiations of their i-Blurry setting, *including the disjoint and Blurry setting*, on the CIFAR-10, CIFAR-100 and TinyImagenet datasets. An ablation study shows the main benefits of the proposed method come from the sample importance based memory scheme.

**Summary Of The Review:**

### Basis for the scores

1. The experiments justify most of the information present in the paper. Some additional information regarding the sample importance based memory management scheme is required. Also, some decisions regarding hyperparameters need to justified, given the *i-Blurry* setting is online and the main motivation of the paper is to propose and solve a more practical CL setting.

2. I'm not sure about the novelty of the sample-importance based memory management scheme and a discussion regarding prior work related to sample-importance is required. The proposed setting on *i-Blurry* is marginally novel as it is a generalization of the disjoint and blurry configurations. Any-time inference and $A_{AUC}$ are  likely novel.

3. Empirical results shown cannot be easily (and extensively) compared to some prior work as there isn't a strict adherence to prior dataset and hyperparameter choices. Some justification for the decisions is required.

4. Insights into the proposed CLIB method along with a nuanced discussion regarding the differences between the algorithms is lacking. There is no description regarding efficiency and scalability.

## Update
The authors clarify almost all of my questions pertaining to the paper. The paper now contains experiments on Imagenet-1k with additional insights into the algorithm and its efficiency. As such, I've improved my scores.

---

> ### Author Response · Authors · 2021-11-13
> **Answer to the questions of Reviewer 6BAu (3/3)**
>
> > **Suggestion** (not a weakness): The paper can also report the performance on some (soft) upper bounds. For instance, what if the task was not restricted to be online. What is the performance when the entire training data is available at once (non CL scenario)?
>
> $\to$ Thank you for the helpful suggestion. We are conducting the ‘joint training’ (non CL) experiments to show the soft upper bound. It will be ready in 2-3 days.
>
> > **There are some typos.**
>
> $\to$ Thank you for pointing them out. We have fixed them in the revision.
>
> > **(Basis for the scores) 2-B. The proposed setting on i-Blurry is marginally novel as it is a generalization of the disjoint and blurry configurations.**
>
> $\to$ We respectfully argue that the generalization is of importance to the research community since it alleviates concerns present in existing CL setups and introduces a more realistic CL setup (it is also acknowledged by **Reviewer F4rS**; ‘an important research direction’).
>
> > **(Basis for the scores) 4-A. … along with a nuanced discussion regarding the differences between the algorithms is lacking.**
>
> $\to$ For Baseline-ER, it performs well as ER is a strong baseline for online CL in general. For EWC++, there is little difference to Baseline-ER as the addition of the episodic memory renders the regularization of EWC++ unnecessary. For BiC, while it shows strong performance in the disjoint setup, it falls short in the blurry and i-Blurry setups as the bias correction in BiC is not well suited for blurry classes. For ER-MIR, it performs similar to the Baseline-ER but scales poorly to larger datasets possibly due to its  memory usage (MIR) not being effective for larger datasets. For RM (Bang et al., 2021), the two-stage training method leads to low any-time inference performance. For GDumb (Prabhu et al. 2020), it performs poorly on later tasks because it trains from scratch at every inference query. The discussions regarding RM and GDumb can be found in Sec. 5.1. We have added more discussion of other methods including RM and GDumb in Sec. A.9 in the revision.

---

> ### Author Response · Authors · 2021-11-13
> **Answer to the questions of Reviewer 6BAu (2/3)**
>
> > **3-B: A note about the efficiency/runtime needs to be specified.**
> > > **(Basis for the scores) 4-B. description regarding efficiency and scalability.**
>
> $\to$ The computational complexity of our method is $O(M)$ where $M$ is the fixed memory size (see line 2 of Alg. 1). Thus, the algorithm can scale to larger datasets as the memory size is kept constant regardless of the data size. Note that for further efficiency, we can use the sample wise importance update described in Alg. 1 at every $k$ training iterations instead of every training iteration. We add this in Sec. A.3 of the revision.
>
> > **4: No information as to why TinyImagenet is challenging for all methods is provided.** There is also no justification provided for choosing TinyImagenet over Imagenet (RM, BiC and GDumb all show their results on Imagenet).
> > > **(Basis for the scores) 3. Empirical results shown cannot be easily (and extensively) compared to some prior work as there isn't a strict adherence to prior dataset...**
>
> $\to$ For CL methods with an episodic memory, the size of the episodic memory with respect to the number of classes (*i.e.,* samples per class stored in the memory) is an important factor for performance. Although TinyImageNet is smaller than ImageNet, TinyImageNet has 200 classes and is equipped with a memory of size 4,000 while ImageNet-1k has 1,000 classes with a memory of size 20,000. Thus, the number of samples per class stored in the memory for training is identical for both datasets. However, TinyImageNet has lower resolution images than ImageNet, which makes learning visual information challenging. Thus, we thought TinyImageNet would be a challenging dataset for our i-Blurry setup due to the image size. Our empirical results show that the accuracy by the best performing method is around 20%’s. Given the already low performance on TinyImageNet and the huge computation cost of conducting ImageNet experiments, we used TinyImageNet. Nevertheless, we do understand that ImageNet-1k experiments provide a lot of value to our submission as a large-scale experiment and are conducting experiments with ImageNet-1k. The results should be ready in about 6-7 days.
>
> > **5: Since the i-Blurry setting is an online one, the choice of hyperparameters are likely to affect the results greatly**. As such hyperparameter choices (updates per sample, memory size and batchsize) need to be justified. Specifically, were the hyperparameters chosen consistent with previous work?
> > > **(Basis for the scores) 1-B. Some decisions regarding hyperparameters need to justified, given the *i-Blurry* setting is online and the main motivation of the paper is to propose and solve a more practical CL setting.**
> > > **(Basis for the scores) 3-A. Empirical results shown cannot be easily (and extensively) compared to some prior work as there isn't a strict adherence to ... hyperparameter choices. Some justification for the decisions is required.**
>
> $\to$ Yes, the chosen hyperparameters (*i.e.*, updates per sample, memory size, and batch size) are consistent with previous works such as RM (Bang et al., 2021) and GDumb (Prabhu et al. 2020). We clarify this under the ‘Implementation Details’ in the ‘Experiment’ sections in the revision.
>
> > **6: For a fixed M, is it reasonable to expect that performance of all algorithms would increase (in general) with increasing N?** This can be used to justify choosing N to be from 0,50,100 (would also be helpful reference for future works).
>
> $\to$ Yes, it is reasonable. The results from top of Table 2 show that the performance increases as $N$, the ratio of disjoint classes, goes from 0 to 100 for all methods. It is because the disjoint setup has about half of the classes for evaluation when averaged overtime, than that of the blurry setup. Thus, it is easier to achieve higher accuracy when $N$ is large. In more details, when there are a total of $k$ classes in a dataset, in a disjoint split the number of encountered classes grows from $0$ to $k$, whereas in a blurry split the number of encountered classes is $k$ after a very short period of time (all classes are encountered very soon in the blurry split). Since we evaluate methods on the classes the model has encountered, in a disjoint split the model is evaluated on $k/2$ classes on average, while in a blurry split the model is evaluated on roughly $k$ classes on average. Therefore, the performance of algorithms will generally increase with larger $N$. Thus, we choose N to be 0, 50, 100 to investigate the impact on various algorithms in Table 2 as you pointed out.

---

> ### Author Response · Authors · 2021-11-13
> **Answer to the questions of Reviewer 6BAu (1/3)**
>
> > **(Weakness) 1: I am unsure if the proposed approach of obtaining sample importance is entirely novel.** A discussion regarding this is necessary as it is likely that similar ideas present (beyond CL for instance) in literature.
> > > **(Basis for the scores) 1. ... Some additional information regarding the sample importance based memory management scheme is required.**
> > > **(Basis for the scores) 2. I'm not sure about the novelty of the sample-importance based memory management scheme and a discussion regarding prior work related to sample-importance is required.**
>
> $\to$ The idea of utilizing sample-wise importance scores in learning better models has been studied for many years in various contexts [A,B,C,D,E; to name a few]. But as each different problem requires different constraints, the idea should be differently rendered to meet the constraints of each task. Our contribution lies in applying this idea to *a new realistic CL context in a novel way* (Please note that **Reviewer Vfw2** acknowledges this by ‘The memory management method is novel’). We have added this discussion in Sec. 4.2 of the revision.
>
> For the prior works in the CL literature, there are some related works such as MIR (Aljundi et al., 2019) and EWC (Kirkpatrick et al., 2017). While MIR also uses a kind of importance scores, MIR and our method are different in the following ways:
> - MIR uses importance scores to select samples from a *constructed memory*
>     - In contrast, we use the sample importance for *constructing* the memory
> - MIR allocates high importance scores to samples that exhibit large *increases* in its loss when the current *streamed data* are used for training.
>     - In contrast, we set high importance scores to samples that incur large *decreases* in *memory’s total loss* when that sample is used for training.
> - MIR’s importance score is calculated only for a single iteration *i.e.,* new importance scores are calculated every iteration.
>     - In contrast, we accumulate the computed importance scores over time to obtain the overall sample-wise importance scores.
> We added more discussions in Sec. A.9.4 of the revision.
>
> EWC is less similar to our method than MIR as it is an offline CL method. It uses the Fisher information matrix as importance scores for model *parameters*. They regularize the neural network’s parameters to be similar to the values of the model of the previous task, with the *parameter-wise* importance scores as their weighting factors for regularization. In contrast, ours is an online CL method that uses *sample-wise* importance scores calculated using loss decreases from training with samples from the memory.
>
> [A] T. Kloek, H. K. van Dijk, "Bayesian Estimates of Equation System Parameters: An Application of Integration by Monte Carlo". Econometrica 46 (1): 1-19, 1978
> [B] Y. Le Cun et al., “Optimal Brain Damage,” NIPS 1989
> [C] H.-S. Chang et al., “Active Bias: Training More Accurate Neural Networks by Emphasizing High Variance Samples,” NeurIPS 2017
> [D] A. Katharopoulos and F. Fleuret, “Not All Samples Are Created Equal: Deep Learning with Importance Sampling,” ICLR 2018
> [E] D. Csiba and P. Richtarik, “Importance Sampling for Minibatches,” JMLR 2018
>
>
> > **(Weakness) 2: A natural language description of Theorem 1 is missing.**
>
> $\to$ Theorem 1 is about what sample to be removed from the fixed sized memory when new samples arrive and are added to the memory. We select the ones that incur the least loss decrease as they contribute less in training the model. We added this in Sec 4.2 in the revision.
>
> > **(Weakness) 3-A: I felt that more insights into the calculation of importance are required in the main text.**
>
> $\to$ We use the following insights in calculating the importance scores of the samples in the memory. (Brief recap of the process of computing the loss difference: the model is trained with a batch randomly drawn from the memory. Then, for all samples in the appended memory ($C$ in Thm. 1), we compute the loss difference *before* and *after* training with the batch.) If the loss decreases, the importance scores of the samples in the batch increase and *vice versa*. In addition, because the importance score of a sample is a relative quantity to that of other samples in the memory, when the importance scores of the samples in the batch decrease, the scores of the samples *not* in the batch increase in comparison. Thus, there are two types of samples that have high importance scores; 1) samples that were in the batch when the loss decreased and 2) samples that were not in the batch when the loss increased. We have added the insights to Sec. 4.2 in the revision. If you need more clarification on the specifics of the insights, please let us know.

---

> ### Author Response · Authors · 2021-11-23
> **Thank you to Reviewer 6BAu**
>
> We are happy that most of the concerns were clarified and we express our thanks for increasing the rating.

---

### Official Review · Reviewer_QQWa · 2021-11-02

**Correctness:** 2
**Technical Novelty And Significance:** 2
**Empirical Novelty And Significance:** 2
**Recommendation:** 3
**Confidence:** 4

**Main Review:**

The paper has excellent plots for the new problem setup. However, without further clarification of the following concepts/comments, the significance of the new setup could be weak.
* **The paper claimed the new setup is both “task-free” and “class-incremental.”** In my opinion, the two setups are not compatible with each other. Specifically, they are different at the output layer in supervised learning.
* **“Any-time inference” may require more rigorous justifications; so as to the new metric.** Only one baseline demonstrated the incapability of making any-time inference in Table 1. The significance of any-time inference needs more evidence.
* **Thm. 1 may suffer from catastrophic forgetting.** Thm. 1 can be seen as a performance-driven memory management strategy. However, it has nothing to do with catastrophic forgetting. In Fig. 4, the proposed method has a projection of performance drop in the later phase. In the long run, it is hard to tell whether the proposed method will outperform the baselines. This performance drop could be attributed to the catastrophic forgetting of Thm. 1.
* **The paper doesn’t discuss the algorithm complexity**, which could be O(M) in general, where M is the memory size.


**Summary Of The Paper:**

The paper proposes a new problem setup in continual learning. As the title suggests, the paper focuses on online, task-free, class incremental, task blurry learning with any-time inference. The authors also came up with new baselines and importance-based memory management. They empirically tested their methods in the proposed problem setup.

**Summary Of The Review:**

There are unsolved concerns about the concepts/comments in the current version. Without addressing them, the proposed problem setup and method could not be that significant for continual learning.

---

> ### Author Response · Authors · 2021-11-13
> **Answer to the questions of Reviewer QQWa (2/2)**
>
> > **Thm. 1 may suffer from catastrophic forgetting**: In Fig. 4, the proposed method has a projection of performance drop in the later phase. In the long run, it is hard to tell whether the proposed method will outperform the baselines.
>
> $\to$ This is an interesting concern and we are conducting experiments for longer task sequences. The results will be ready in 2-3 days.
>
> > **Thm. 1 may suffer from catastrophic forgetting**: it has nothing to do with catastrophic forgetting.
>
> $\to$ Yes, Thm. 1 does not explicitly address the catastrophic forgetting. Note that performance of CL methods is determined by the trade-off between the stability-plasticity [A] (*i.e.*, forgetting and intransigence (Chaudhry et al., 2018)). By Thm. 1, we focus more on the plasticity in the trade-off, *i.e.*, we aim to learn new data quickly, by always adding the newly encountered sample to the memory. As a result, our method reaches high accuracy much quicker than other methods (compare the curve by our method to curves by other methods in Fig. 4). More importantly, our choice of focusing on the plasticity over stability results in better performance than other CL methods by large margins in both existing and the proposed CL scenarios, implying better trade off between the stability-plasticity.
>
> Furthermore, while we do not explicitly address the forgetting in Thm. 1, it is interesting to note that the forgetting performance ($F_5$) shown in Table 7 in Sec. A.10 of our method is comparable to most other methods, except for two outliers (BiC and RM). We believe it is because we use an episodic memory in our memory-only training as it is known that episodic memory mitigates catastrophic forgetting (Rebuffi et al., 2017).
>
> [A] Mermillod, M., Bugaiska, A., & Bonin, P. (2013). The stability-plasticity dilemma: Investigating the continuum from catastrophic forgetting to age-limited learning effects. Frontiers in psychology, 4, 504.
>
> > **This performance drop could be attributed to the catastrophic forgetting of Thm. 1.**
>
> $\to$ We cordially argue that the “projection of performance drop” is attributed more to the increasing number of encountered classes at later phases in the i-Blurry setup than to catastrophic forgetting of Thm 1. To investigate the accuracy when the number of encountered classes **do not** increase at later phases (*i.e.*, no new class is encountered after the first task), we show the accuracy-to-{# of samples} curve for the *blurry setup* results in Table 2 for $N=0$ in Fig. 5 in Sec. A.11 of the revision. As shown in the figure, there is little performance drop at later phases for all methods, which supports our argument that the drop is mostly due to the increasing number of classes over time.
>
>
> > **The paper doesn’t discuss the algorithm complexity, which could be O(M) in general, where M is the memory size.**
>
> $\to$ Yes, the complexity of our algorithm (Alg. 1) is $O(M)$, where $M$ is the size of episodic memory, *i.e.*, a constant. Note that for better practical efficiency, we can use the sample-wise importance update described in Alg. 1 every $k$ iterations instead of every iteration. We add this to Sec. A.3 in the revision.

---

> > ### Comment · Reviewer_QQWa · 2021-11-21
> > **Thanks for the authors feedback**
> >
> > Thanks for the authors' efforts of clarifying Thm. 1.
> >
> > The plasticity-stability trade-off sounds like a good explanation and helps readers understand the implications of Thm 1.
> >
> > To test whether there is catastrophic forgetting, the author can conduct a simple experiment: first, update the model on one image class, and then update the model on other image classes for a while. Afterward, check the model's performance on the un-seen class to see if there is catastrophic forgetting.
> >
> > The experiment in Sec. A. 11 looks that it doesn't directly test the catastrophic forgetting problem and maybe off the goal.

---

> > > ### Author Response · Authors · 2021-11-22
> > > **Clarification on the experiment in Sec. A. 11**
> > >
> > > > To test whether there is catastrophic forgetting, the author can conduct a simple experiment: first, update the model on one image class, and then update the model on other image classes for a while. Afterward, check the model's performance on the un-seen class to see if there is catastrophic forgetting.
> > >
> > > > The experiment in Sec. A. 11 looks that it doesn't directly test the catastrophic forgetting problem and maybe off the goal.
> > >
> > > $\to$ Please note that the experiment in Sec. A.11 was to test if there is a performance drop when no new classes are introduced after the first task (i.e. blurry setup) and not to see if there was catastrophic forgetting. We have clarified this in the revision. We observe that in the blurry setup there is no performance drop for all the tested methods, which supports our claims that the performance degradation may be caused by new classes being encountered as tasks progress.
> > >
> > > While the proposed experimental protocol is interesting and we considered adding the results, there were two reasons why we decided not to. First, there is already a well-established metric for catastrophic forgetting (the $F_\text{last}$ measure from Chaudhry et al., 2018, previously denoted as $F_5$ in the last revision) and we show the results with the $F_\text{last}$ measures in Sec. A.10 to compare catastrophic forgetting amongst methods. As noted in our last reply, our method has similar $F_\text{last}$ performance to most compared methods.
> > >
> > > Second, the proposed experiment forces a way of sampling classes for training, which means that CL methods that use an episodic memory and use it to sample data for training (e.g., all the methods compared in our submission) would need to have a portion of their method replaced to varying degrees with some methods sacrificing more of their method than others. For example, ER-MIR proposes a specific way of sampling data from the memory for training. Replacing the proposed way with the above experimental protocol would result in almost disabling ER-MIR’s proposed methodology altogether. We believe that as a forgetting measure ($F_\text{last}$ measure) already exists, unevenly replacing portions of CL methods for comparison would be unfair.

---

> ### Author Response · Authors · 2021-11-13
> **Answer to the questions of Reviewer QQWa (1/2)**
>
> Thank you for the encouraging remarks on the new problem set-up. We clarify your comments as follows.
>
> > **“task-free” and “class-incremental.” are not compatible with each other in my opinion**. Specifically, they are different at the output layer in supervised learning.
>
> $\to$ We first clarify the definition of both terms and respectfully argue that they are compatible. We follow the definition of “task-free” in (Aljundi et al., 2019b), and the definition of “class-incremental” in (Prabhu et al., 2020, Bang et al., 2021). Specifically, ‘task-free’ refers to task boundaries (or task-id) being unknown to the model during the training phase, so the output layer of a model is a classifier for all the classes encountered so far. ‘Class-incremental’ refers to task-id’s being not available at inference time, so as in task-free, the last layer is a classifier for all the classes encountered so far. Thus, the ‘task-free class-incremental’ setup refers to the task information not being provided in both training and inference phases where the output layer is a classifier for all the encountered classes.  Additionally, please note that GMED (X. Jin et al., 2020) and Class-Incremental Learning with Generative Classifiers (G. van de Ven et al., 2021) also describe their set-up as both task-free and class-incremental. Also note that the **Reviewer Vfw2** implicitly acknowledges the combination is possible by “Evaluation is carried out in the class incremental, online, task-free setting, which is a good decision in my opinion.”
>
>
> > **More rigorous justifications on “Any-time inference” and the new metric.** Only one baseline demonstrated the incapability of making any-time inference in Table 1.
>
> $\to$ Any-time inference could be justified for practical online CL by following reasons. First, in real world online CL scenarios, it is reasonable to assume that we do not know when the inference query is provided because inference queries would occur sporadically, not only at fixed task boundaries as conventional CL set-ups, *e.g.*, offline CL, assume. Please note that **Reviewer 6BAu** also acknowledges this by *This setting is also online and thus the paper is interested in continuous model evaluation i.e., any-time inference too.* Second, any-time inference evaluates the accuracy continuously at a higher frequency, which is helpful for investigating how quickly a CL method learns information from new data.
>
> To properly evaluate the performance of any-time inference, we propose a new metric of evaluating models continuously, called $A_{AUC}$. $A_{AUC}$ calculates the area under the accuracy curve which captures the model performance better for sporadic inference queries rather than evaluating only at the task boundaries. By $A_{AUC}$, we can observe that RM (Bang et al. 2021), the state-of-the-art online CL method on the realistic blurry CL setup, could perform disappointingly in this more realistic inference scenario (due to its two-stage training). In addition, as RM establishes the new state of the art in the blurry setup, it is reasonable to expect that new methods may adopt RM’s two stage training. To promote the future research towards realistic setup, we respectfully argue that metrics like $A_{AUC}$ are important (*i.e.*, promote developing methods other than the two stage training scheme).

---

> > ### Comment · Reviewer_QQWa · 2021-11-21
> > **The author may have a misunderstanding of task-free continual learning and exaggerate their contribution**
> >
> > > Specifically, ‘task-free’ refers to task boundaries (or task-id) being unknown to the model during the training phase, so the output layer of a model is a classifier for all the classes encountered so far. ‘Class-incremental’ refers to task-id’s being not available at inference time, so as in task-free, the last layer is a classifier for all the classes encountered so far.
> >
> > The author may misunderstand the task-free setup, which shouldn't be mixed up with the class-incremental setup. For task-free setup, each task's id is unknown both in training and testing time, and each task doesn't share the label space with others. Thus in testing time, people must infer each task's id and select the corresponding set of output layers before making predictions. On the other hand, the class-incremental setup has a shared label space across all time steps, as described in the paper. People don't have to infer the task's id. For a concrete definition of task-free or task-agnostic setup and various definitions, please see pages 28-36 of the CL tutorial [https://icml.cc/media/icml-2021/Slides/10833_njzbXvu.pdf]
> >
> > As the author mentioned, they get their definition from one of the first task-free works, Aljundi et al., 2019. Aljundi et al., 2019 do distinguish their setup from class-incremental setup. See the 3rd paragraph in Section 5.
> >
> > That said, the authors can have a different understanding of the task-free setup. But the setup should be well-defined and distinguished from other setups, i.e., class-incremental setup, for which I failed to see. Otherwise, the paper's contribution could be exaggerated.
> >
> > [Aljundi et al., Task-Free Continual Learning, CVPR 2019]
> >
> > > More rigorous justifications on “Any-time inference” and the new metric.
> >
> > Thanks for the author's clarification. Here I want to make some clarifications for my previous wording. In my first round review, I mentioned only one baseline shows incapability in the online learning setup. I didn't mean the"anytime inference" is not practical. Instead, I intended to understand why the paper makes "anytime inference" as a selling point. In my understanding, the "anytime inference" is an intrinsic evaluation strategy for online continual learning, like the step-ahead evaluation.
> >
> > Fig. 2 shows the baseline RM is not capable of doing online learning, which could be used to justify that the paper contributes, e.g., an online learning algorithm over RM or something else. But it is not a good reason to say they contribute a new "anytime inference" evaluation framework. This evaluation framework, by my understanding, is a must for online learning setup.

---

> > > ### Comment · Reviewer_Vfw2 · 2021-11-22
> > > **The task-free setting**
> > >
> > > I must respectfully disagree with my fellow reviewer wrt the task-free setting. The task-free setting, as I understand it, is perfectly compatible with class-incremental learning in the right problem context. I think the issue here is that "task" means different things with different problems. In particular, the reviewer writes that "each task doesn't share the label space with others". But consider the case of split MNIST. Here, the different tasks can be considered as the digit subsets {0, 1}, {2, 3}, ... In the multi-head (task incremental) setting, each task indeed has its own (binary) label space. But in the (harder) class incremental setting, each task has a common label space, which is all ten digits. In the task-free version of this, the system is not told when one such task ends and the next begins. In this setting, the task boundaries can also be blurred, as in the authors' paper. Perhaps the authors can confirm whether this is also their understanding?

---

> > > > ### Comment · Reviewer_QQWa · 2021-11-22
> > > > **Thanks for the discussion**
> > > >
> > > > Thanks for initiating this discussion. Yes, I wrote, "For task-free setup ... each task doesn't share the label space with others." But I didn't say other problem setups, e.g., task-incremental setup as you mentioned, don't have this feature. My goal was to distinguish task-free setup from class-incremental setup. In terms of the task-free setup, like the following, I will try to distinguish "task-increment setting" from "task-free setting". If my clarification misses important points, feel free to correct me.
> > > >
> > > > I agree that the split MNIST is a good example for clarifying the concepts. Since you mentioned the task incremental setting as in "...digit subsets {0, 1}, {2, 3}, ... In the multi-head (**task incremental**) setting", my understanding is that task-incremental setting is *not* task-free setting. Here is my reasoning. If you know the data set changes from {0,1} to {2,3} and the labels changes to {2,3}, then you know the task boundaries, which is no longer task-free to task-agnostic. Besides, {0,1} and {2,3} also share the label space, although they are multi-head. Another evidence for the fact that the task-incremental setting assumes known task boundaries is the definition on page 30 of the CL tutorial (https://icml.cc/media/icml-2021/Slides/10833_njzbXvu.pdf). An exception is tasks like {0,1} and {1,2} where different tasks have their own label spaces and you don't want to mix them up.
> > > >
> > > > On the other hand, the task-free or task-agnostic setting is much more difficult. In the example of split MNIST, the task-agnostic setting says you don't know the digit labels at all. Even though the input digits changes from {0,1} to {2,3}, the labels are still just *binary*. It is the agent's responsibility to infer if the input changes and whether to initiate a new task training procedure or not. During test time, given an input sample, the agent must select the correct output set and make a prediction. Note, you can say the labels depending on the inputs and thus the inputs split a (shared) label space, but the task-specific output is different from the class-increment setup.
> > > >
> > > > If you and the paper's authors are referring to the task-incremental setup, I think it should be made clear and separated from a task-free setup where the task boundaries are **unknown**.

---

> > > > > ### Comment · Reviewer_Vfw2 · 2021-11-22
> > > > > **Task-free setting**
> > > > >
> > > > > You wrote "In the multi-head (task incremental) setting", my understanding is that task-incremental setting is not task-free setting". Yes, exactly. My point was to contrast this with the class incremental setting (single head, all labels), where it *does* make sense to talk about class incremental learning.
> > > > >
> > > > > You wrote: "the task-agnostic setting says you don't know the digit labels at all. Even though the input digits changes from {0,1} to {2,3}, the labels are still just binary." This is not what I take to be the class incremental setting with split MNIST, where each task has a common label space, which is all ten digits (even though only each task only involves a subset of those labels).
> > > > >
> > > > > Anyway, I should let the authors continue this.
> > > > >
> > > > > (We can probably all agree this is a sign of how confusing the field of CL is, with so many variations in setup, and differet authors using the same term to mean different things.)

---

> > > > ### Author Response · Authors · 2021-11-23
> > > > **Reply to question of Reviewer Vfw2**
> > > >
> > > > We again thank you for participating in this helpful discussion. Your understanding is precisely what we intended when we described our problem setup. However, as revealed in the discussion with **Reviewer QQWa**, the terminology can be differently interpreted in different contexts. Thus, we provided additional clarification in our reply to **Reviewer QQWa.** If you have any further comments, feel free to let us know. Thank you!

---

> > > ### Author Response · Authors · 2021-11-22
> > > **Clarification of our setup**
> > >
> > > > The author may misunderstand the task-free setup, which shouldn't be mixed up with the class-incremental setup.
> > > > - For task-free setup, each task's id is unknown both in training and testing time, and each task doesn't share the label space with others.
> > > >    - Thus in testing time, people must infer each task's id and select the corresponding set of output layers before making predictions.
> > > > - On the other hand, the class-incremental setup has a shared label space across all time steps, as described in the paper. People don't have to infer the task's id.
> > > > - For a concrete definition of task-free or task-agnostic setup and various definitions, please see pages 28-36 of the CL tutorial [https://icml.cc/media/icml-2021/Slides/10833_njzbXvu.pdf]
> > > > As the author mentioned, they get their definition from one of the first task-free works, Aljundi et al., 2019. Aljundi et al., 2019 do distinguish their setup from class-incremental setup. See the 3rd paragraph in Section 5.
> > > > That said, the authors can have a different understanding of the task-free setup. But the setup should be well-defined and distinguished from other setups, i.e., class-incremental setup, for which I failed to see. Otherwise, the paper's contribution could be exaggerated.
> > >
> > > $\to$ We thank the two reviewers (**QQWa** and **Vfw2**) for their active and constructive discussion on clarifying CL terminology. We cordially note that in the 3rd paragraph of Sec. 5 in (Aljundi et al., 2019) (referred by **Reviewer QQWa**), they mention **applying their task-free setup to the class-incremental setting as a potential future work**, which is what we have done with our setup. Following Aljundi et al., 2019, we used the term “task-free class-incremental” for referring to a CL setup where the classes are added incrementally and the task-id is **not** used during both training or testing. Going one step further, we also introduced blurry task boundaries to our setup, such that the incrementally added classes are not restricted to be disjoint. In sum, our goal is to introduce a more realistic and practical CL setup by proposing CL with incrementally added classes that may be blurry and one that also does not use the task-id at all (*i.e.*, both in training and testing).
> > >
> > >
> > > We agree with **Reviewer QQWa** that task-free cannot be class-incremental based on the terminology from the ICML Tutorial. It seems that our work is difficult to categorize using terminology from the ICML Tutorial while not being difficult to categorize using terminology from Aljundi et al., 2019. This implies that the currently defined CL terminology may be insufficient to describe recently emerging CL setups that are more realistic and practical including ours. In this sense, we also agree with **Reviewer Vfw2** that the terminology used in CL needs a more rigorous definition to improve the field of CL and leave that as future work.
> > >
> > >
> > > If it is deemed that using existing CL terminology to describe our new realistic and practical CL setup causes more confusion than it helps, we are willing to change the description such that it is clearer to the reader.
> > >
> > >
> > > In the meantime, we cordially remind the reviewers (**QQWa** and **Vfw2**) that we were focused on introducing a more realistic and practical setup and clarify it one last time:
> > > Our setup follows class-incremental setup (*i.e.*, incrementally added classes) but **does not** use task ids in training time unlike the typical class incremental setup. That is why we call it “task-free class-incremental.”
> > > Our setup also has blurry task configuration; the incrementally added classes in tasks are not restricted to be disjoint. That is why we call it “blurry” task configuration.
> > > Note that existing setups such as the blurry setup are **not** class-incremental.
> > >
> > >
> > > Finally, to further our understanding of CL, we would greatly appreciate it if **Reviewer QQWa** could point us to any references that mentions “task-free” as not being able to share the label space as its definition or constraint.

---

> > > > ### Comment · Reviewer_QQWa · 2021-11-22
> > > > **Thanks for the response**
> > > >
> > > > Thanks for all the valuable discussion. Like the following, I'm trying to reconcile the differences and proceed with the work. I suggest dropping the word "task" and focusing on the class-incremental setup.
> > > >
> > > > I respect that and understand that you are guarding your point. It seems the usage of the word  "task" causes some confusion, especially you mention your "task-free" means there are no other tasks at all. People may interpret a "task-free" method to be the one applicable to classification tasks, regression tasks, or even others without known task boundaries. Thus, to reconcile the differences, I suggest focusing on the class-incremental setup as the main part with a blurry or changing class distribution over time. This can avoid other potential misinterpretations for "task-incremental" and "task-free."
> > > >
> > > > As for the issue of "shared label space," I give it as a way to contrast task-based setups and class-incremental setups.

---

> > > > > ### Author Response · Authors · 2021-11-23
> > > > > **Response to terminology ‘Task’ could be confusing.**
> > > > >
> > > > > Thank you for your suggestion. We agree with you that the term **task** might cause confusion because different works define it in various ways. To further clarify our setup, we will describe our setup as a **blurry class incremental setup without using task id** and revise our paper in the final version.

---

> > > ### Author Response · Authors · 2021-11-22
> > > **Clarification on concerns regarding anytime inference**
> > >
> > > > Thanks for the author's clarification. Here I want to make some clarifications for my previous wording. In my first round review, I mentioned only one baseline shows incapability in the online learning setup. I didn't mean the "anytime inference" is not practical. Instead, I intended to understand why the paper makes "anytime inference" as a selling point. In my understanding, the "anytime inference" is an intrinsic evaluation strategy for online continual learning, like the step-ahead evaluation.
> > >
> > > > Fig. 2 shows the baseline RM is not capable of doing online learning, which could be used to justify that the paper contributes, e.g., an online learning algorithm over RM or something else. But it is not a good reason to say they contribute a new "anytime inference" evaluation framework. This evaluation framework, by my understanding, is a must for online learning setup.
> > >
> > > $\to$ We agree that an “anytime inference” evaluation framework is a must for online learning setups. However, we cordially note that there is little previous work for online CL that actually uses the “anytime inference” evaluation framework or metrics that could accurately evaluate anytime inference performance. Most if not all online CL evaluate only at the discrete task boundaries (Aljundi et al., 2019, Prabhu et al., 2020, Bang et al., 2021 and many more), which is ill-suited to test anytime inference. To mitigate these shortcomings, we proposed a new metric to measure anytime inference performance such that the seemingly must evaluation framework for online learning setups could actually be used in future online CL. Thus, we argue that  our new metric to promote the anytime inference evaluation framework can contribute to the online CL community.

---

### Official Review · Reviewer_F4rS · 2021-11-02

**Correctness:** 3
**Technical Novelty And Significance:** 2
**Empirical Novelty And Significance:** 3
**Recommendation:** 8
**Confidence:** 4

**Main Review:**

### Strengths

- The proposed benchmark protocol, “i-Blurry”, is reasonable and interesting. It is important to analyse the results when some of the classes are not disjoint in continual learning tasks.

- The authors provided extensive experimental results.

- The proposed method is technically sound and proven to be effective by the empirical results.

- The paper is well-organized and easy to follow.

&nbsp;

### Weaknesses

- ***The technical contributions of this paper are somewhat weak.*** In this paper, the authors mainly propose three components in the method section: “sample importance memory”, “memory-only training”, and “adaptive LR scheduling”. For the first component, the main idea is to find the optimal exemplars that can minimise the total loss so that it could be regarded as a simplified version of “mnemonics” (Liu et al., 2020). For the second component, it can be summarised as “sample importance memory”+GDumb (Prabhu et al. 2020). For the third component, adaptive LR can benefit all classification models. The authors don’t explain why it is important in continual learning tasks.

-  ***This paper is not self-contained.*** Alg.3 and Section A.3 are not some additional information and results. I cannot understand the proposed method without reading Alg.3 and Section A.3. Thus the authors should definitely include these parts in the main paper.

- ***The authors only provide experiment results on small-scale datasets.*** Most continual learning papers, such as iCaRL (Rebuffi et al., 2017) and BiC (Wu et al., 2019), provide the results on large-scale datasets (e.g., ImageNet-1k). As the authors trying to establish a new benchmark protocol, it is important to provide the results on large-scale datasets.

- ***An ablation study on the number of tasks should be provided.*** In many continual learning papers (iCaRL (Rebuffi et al., 2017), BiC (Wu et al., 2019), etc.), the number of tasks will significantly influence the continual learning performance. However, I don’t find an ablation study on the number of tasks in this paper. I don’t even see an explanation about how the authors choose that hyperparameter.

**Summary Of The Paper:**

In this paper, the authors proposed a new benchmark protocol (i-Blurry) for continual learning. In this benchmark protocol, the class distribution is class incremental and has blurry task boundaries, and the training is online. They also propose a new method, CILB. This method contains three important components: “sample importance memory”, “memory-only training”, and “adaptive LR scheduling.” Extensive experimental results are provided to show the effectiveness of the proposed method.

**Summary Of The Review:**

Overall, I think this is an interesting paper. The authors design a new benchmark for continual learning and propose a simple yet effective method. My primary concern is that “how likely is it for following researchers to refer to this benchmark?” As there are already many different benchmark protocols in continual learning, I think the following researchers will tend to choose the benchmark protocol already used in many popular papers. Nevertheless, I still think the paper might be useful for the continual learning community.

My rating is “borderline accept”, and I will consider upgrading my rating if the authors successfully address my questions.

&nbsp;

### Post-rebuttal update
The authors addressed most of my concerns in the rebuttal. They also provided the results I asked for in the revision (e.g., the results on ImageNet-1k). I think I tend to accept this submission. So I upgrade my rating to eight.

---

> ### Author Response · Authors · 2021-11-12
> **Answer to the question of Reviewer F4rS**
>
> > **First component of technical contributions**: sample importance memory could be regarded as a simplified version of “mnemonics” (Liu et al., 2020)
>
> $\to$ Please note that utilizing the total loss in learning a better model has been studied for many years [A, B]. We argue that our method and ‘Mnemonics’ (Liu et al., 2020) use this idea in very different ways. Mnemonics uses the total loss decrease to transform *randomly selected exemplars* and store transformed exemplars in the memory. In contrast, our sample importance memory *selects important* samples by computing sample-wise importance scores from loss decreases.
>
> Additionally, Mnemonics is an offline CL method which needs to store the entire task’s data whereas our sample importance memory is an online CL method which only needs to track the loss decrease for samples in a small memory. Thus, we argue that the proposed sample importance memory is a novel instance of `importance sampling’ method for episodic memory management in online CL setup and not a simplification of Mnemonics.
>
> [A] T. Kloek, H. K. van Dijk,"Bayesian Estimates of Equation System Parameters: An Application of Integration by Monte Carlo". Econometrica 46 (1): 1-19, 1978
> [B] Y. Le Cun et al., “Optimal Brain Damage,” NIPS 1989
>
> > **Second component of technical contributions**: memory-only training can be summarised as “sample importance memory”+GDumb (Prabhu et al. 2020)
>
> $\to$ We respectfully argue that 'memory-only training’ is quite different from ‘sample importance memory + GDumb.’ As GDumb is comprised of greedy sampler + dumb learner where dumb learner trains only using memory, we understand that our method may be viewed as sample importance memory + dumb learner. However, the dumb leaner only trains the model when an inference query occurs not when new samples are encountered. This makes usage of sample importance memory with the dumb learner incompatible because in sample importance memory, the model needs to be trained when new samples arrive (*e.g.*, from the online stream) in order to compute the importance scores. Thus, our method is **not** sample importance memory + dumb learner but rather sample importance memory + memory-only training. In particular, the memory-only training differs from dumb learner in that the dumb learner resets the previous model trains a new model from scratch when an inference query is given. In contrast, our memory-only training continually updates the previously learned model. Because of this, there are two consequences:
> - Dumb learner does not accumulate past knowledge whereas our method can accumulate past knowledge from previous tasks.
> - Dumb learner’s computational cost increases with the number of inference queries because a new model has to be trained from scratch before inference while ours does not.
>
> > **Third component of technical contributions**: why is adaptive LR scheduling important in CL?
>
> $\to$ Because the training data distribution changes in time, the LR to optimize the model has a greater need to adapt to the data distribution in the current time step than conventional classification set-ups where the training data is fixed. We have added this to Sec. 4.4 in the revision.
>
> > **The paper is not self-contained.** Include Alg. 3 and Sec. A.3. to main paper
>
> $\to$ We have added more explanations about adaptive LR in section 4.4 in revision as suggested. Although we could not move Alg. 3 into the main text due to space limitations, the added explanation would supply sufficient information about the algorithm and that Alg. 3 would serve as the summary in the appendix.
>
> > **The authors only provide experiment results on small-scale datasets.** Large scale experiments (*e.g.*, ImageNet-1k)
>
> $\to$ We agree that results on large-scale datasets are important and are conducting experiments with the ImageNet-1k dataset. We will report the results in about 6-7 days.
>
> > **An ablation study on the number of tasks should be provided.** Ablation study & discussion on choice of number of tasks
>
> $\to$ We understand that an ablation study on the number of tasks is a valuable addition and are conducting the experiments. We will report the results on the CIFAR100 dataset in 2-3 days. The number of tasks was chosen to be $5$ to be consistent with previous works such as RM (Bang et al. 2021).

---

> > ### Comment · Reviewer_F4rS · 2021-11-12
> > **Thanks for the authors' feedback**
> >
> > Thanks for the feedback from the authors.
> >
> > The authors addressed most of my concerns. However, one of my concerns (raised in **Summary Of The Review**) is still not answered: “how likely is it for following researchers to refer to this benchmark?” As there are already many different benchmark protocols in continual learning, I think the following researchers will tend to choose the benchmark protocol already used in many popular papers.
> >
> > If the authors successfully address the above question and provide the results they promised in the revision, I am happy to upgrade my rating to eight.
> >
> > &nbsp;
> >
> > Best,
> >
> > Reviewer F4rS

---

> > > ### Author Response · Authors · 2021-11-13
> > > **Answer to the missing question**
> > >
> > > We thank the reviewer F4rS for the encouraging remarks and apologize for the missing question.
> > >
> > > > **"how likely is it for following researchers to refer to this benchmark?"**: As there are already many different benchmark protocols in continual learning, I think the following researchers will tend to choose the benchmark protocol already used in many popular papers.
> > >
> > > $\to$ Thank you for pointing out this important issue. As a matter of course, we plan to upload all dataset splits, codes and trained models for reproducible baselines in a public repository (we have almost completed preparing the code and models). To promote future research on this more realistic continual learning, we will maintain a public leaderboard in a popular platform (*e.g.*, Kaggle) or our own, and publicize the proposed benchmark by hosting challenges in workshops in respectful venues. By collaborating with some industrial partners, we also plan to expand existing datasets or curate a new one with more real-world data in the i-Blurry setup for challenges.
> > >
> > > We will soon add the promised results in the revision. Thank you.

---

> > > > ### Comment · Reviewer_F4rS · 2021-11-13
> > > > **Thanks for the authors' feedback**
> > > >
> > > > Thanks for the additional feedback from the authors.
> > > > I am satisfied with this answer, and I am looking forward to the revision.

---

> > > > > ### Comment · Reviewer_F4rS · 2021-11-22
> > > > > **Thanks for the revision.**
> > > > >
> > > > > Thanks for the revision.
> > > > >
> > > > > I am satisfied with the results, and I have upgraded my rating.

---

> > > > > > ### Author Response · Authors · 2021-11-23
> > > > > > **Thank you to Reviewer F4rS**
> > > > > >
> > > > > > We are glad that you liked our revision and we thank you for increasing the rating.

---

### Author Response · Authors · 2021-11-13
**General response**

We thank the reviewers for their constructive feedback and encouraging remarks such as interesting and reasonable problem setup (F4rS, QQWa), the method being novel and well-performing (Vfw2) across various settings (6BAu), and extensive experiments supporting our claims (F4rS, 6BAu).

To accommodate the additional information added in the revision, some information regarding the baseline has been moved from Sec. 4.1 to the Sec. A.1 in the appendix.

---

### Author Response · Authors · 2021-11-16
**Second revision uploaded**

**Second revision uploaded:**

We have updated the second revision with the following new results:
- Added “Joint Training” results as a soft upper bound in Table 1 (Suggested by Reviewer 6BAu).
- Added ablation studies for  the number of tasks on CIFAR100 in Table 5 in Sec. A.6 (For Reviewer F4rS & QQWa).
  - We vary the number of tasks from 5 to 10 or 25.
  - Our method (CLIB) still outperforms other CL methods by large margins and shows only minor performance drops with longer task sequences.
  - More discussion can be found in Sec. A.6.
- Fixed some more typos

---

### Author Response · Authors · 2021-11-22
**Third revision uploaded**

**Third revision uploaded:**

We have updated the third revision with the following new results:
- Added ImageNet results in Table 1 and Fig 4 (**Reviewer F4rS & 6BAu**).
  - Following BiC (Wu et al., 2019), GDumb (Prabhu et al., 2020), and RM (Bang et al., 2021), we use a single random seed for the results.
  - Our method (CLIB) achieves the best results on $A_\text{AUC}$ and $A_{avg}$.
  - Interestingly, most methods have higher performance on ImageNet than on TinyImageNe. It is aligned with our intuition in our reply to **Reviewer 6BAu**.
  - We have also updated Table 8 in Sec. A.10 to show forgetting measures ($F_{\text{last}}$ from Chaudhry et al., 2018) for the new ImageNet experiments and added new discussion in the respective section (Sec. A.10).
- Fixed some more typos

---

### Decision · Program_Chairs · 2022-01-20

**Decision:**

Accept (Poster)

**Comment:**

The authors propose a new continual-learning setting with a few distinguishing features: 1) the task boundaries are blurry (in other words, past task samples can reappear); 2) training is online; and 3) evaluation using online accuracy (instead of average accuracy). The authors also propose a useful method for this scenario and benchmark it using four different datasets.

The first round of review pointed to two main limitations of the manuscript.
+ The authors only provided small-scale experiments. The reviewers argued that for the setup and method to have an impact having good results using larger-scale data would go a long way.
+ Whether “task-free” and “class-incremental” were compatible.

For the former, the authors were very reactive and provided results using a standard "ImageNet for CL" dataset.

For the latter, I must thank the authors and also the reviewers for discussing this thoroughly. In the end, my understanding is that there was a reconciliation that both were in fact compatible, but the reviewer suggested that this be discussed very clearly by the authors. I second this suggestion. The CL field given its many slightly different settings might be partly to blame here (reviewer Vfw2 made a similar comment, and I also thank them for playing a role in resolving the issue).


A few additional thoughts:
+ I believe that more general setups in CL are worthwhile even in the absence of any immediate applications. This is especially true since some of the standard CL assumptions do not seem to be well motivated. However, I find that claiming that something is more realistic requires grounding (e.g. a set of examples from the "real world" or a specific domain/setting). I know the authors backed some of their claims with references, but different real-world problems will come with different limitations and I would be hesitant to use phrases such as "most real-world" settings without thorough justification.
+ While different from the core of your work, I believe the framework proposed in this other recent paper has similar goals (although the setup allows pre-training and is not online). Might be worth knowing about it in case you do not:
Online Fast Adaptation and Knowledge Accumulation (OSAKA): a New Approach to Continual Learning, NeurIPS 2020
https://papers.nips.cc/paper/2020/file/c0a271bc0ecb776a094786474322cb82-Paper.pdf

All in all, this is a good contribution that proposes an interesting and rich setting along with a good baseline method for it. I strongly encourage the authors to follow through on their promise to provide the community with code, dataset splits, Kaggle leaderboard, etc., as a way to maximize the impact of their work.